# KnowOS: Knowledge-driven Large Language Models for Operating System Kernel Tuning

## Abstract

Operating System (OS) kernel tuning involves systematically optimizing kernel configurations to enhance system performance. Despite recent advancements in large language models (LLMs), kernel tuning remains a significant challenge due to: (1) the semantic gap between abstract tuning objectives and the specific config options, (2) the limited environmental interaction leading to LLM hallucinations, and (3) the rapid evolution of kernel versions. To address these challenges, we introduce KnowOS, a framework powered by knowledge-driven LLMs for automating kernel tuning. KnowOS leverages three key innovations: structured knowledge construction and mapping, knowledge-driven configuration generation, and continuous knowledge maintenance. Extensive experiments demonstrate that KnowOS achieves performance improvements ranging from 7.1% to 155.4% over default configurations across standard OS benchmarks and real-world applications. These results highlight the potential of structured knowledge representations in overcoming the limitations of pure LLM-based approaches for system optimization. Our code is available at `https://anonymous.4open.science/r/KnowOS-B274`.

## 1 Introduction

Operating systems (OS) serve as the essential bridge between hardware and software, acting as the foundation of modern computing systems. The Linux kernel, as the core "brain" of the OS, manages critical hardware resources such as CPU, memory, and I/O for all applications. One of the most effective methods for improving OS performance lies in **kernel tuning** Martin et al. (2021); Evang & Dreibholz (2024). This systematic process involves adjusting kernel configurations to optimize system performance for specific workloads, as illustrated in Figure 1.

However, Kernel tuning remains challenging due to the vast kernel space with over 18,000 config options Jung et al. (2021) and complex dependencies between them Mortara & Collet (2021). Traditional manual tuning methods Tang et al. (2015); Ts'o (2019) rely on expert knowledge, which is time-consuming and labor-intensive. Machine learning-based approaches Acher et al. (2019); Ha & Zhang (2019) are limited by their reliance on extensive training datasets

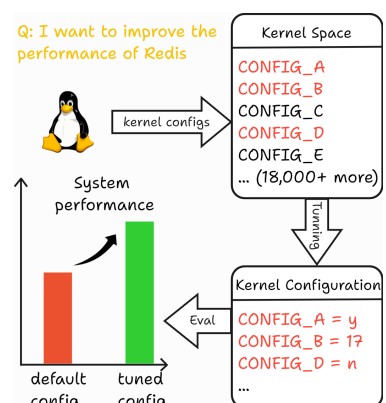

Figure 1: Kernel tuning involves optimizing configurations to enhance system performance for specific workloads.

and struggle with generalization across varying hardware and workloads. Recent advancements in Large Language Models (LLMs) OpenAI (2024); DeepSeek-AI (2025) have demonstrated significant potential for automating kernel tuning tasks Chen et al. (2024), leveraging their extensive knowledge base and natural language processing capabilities.

Despite these advancements, as highlighted in Figure 2, three critical challenges hinder the effective and practical application of LLMs in kernel tuning: **(1) Difficulty in mapping abstract tuning objectives to concrete config options**: LLMs often struggle to align abstract tuning objectives, expressed in natural language, to specific kernel config options, resulting in irrelevant or suboptimal tuning outcomes. **(2) Insufficient environmental interaction leads to LLM hallucinations**: Current

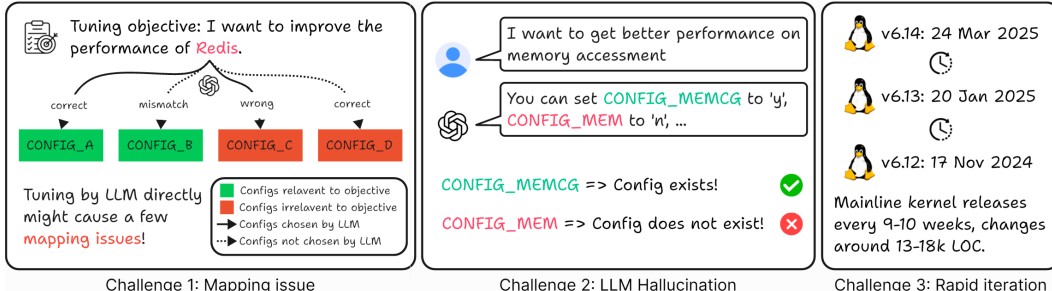

Figure 2: Challenges on OS kernel tuning using LLM. The first challenge is that it is difficult for LLM to map abstract objectives to concrete configuration items. The second challenge is that LLMs may hallucinate, resulting in giving non-existent configurations. The third challenge is the rapid iteration of the kernel configuration, which typically changes every few months.

LLMs lack systematic mechanisms to interact with the complex and vast kernel space, leading to invalid or hallucinated responses Wang et al. (2023). **(3) Rapid kernel iteration and knowledge decay**: The rapid evolution of the Linux kernel (with 13,000 to 18,000 commits per release and new major versions every 2 to 3 months Kroah-Hartman (2019)) outpaces LLMs' ability to maintain up-to-date tuning knowledge.

To overcome these challenges, we introduce **KnowOS**, a novel knowledge-driven framework powered by LLMs for automating OS kernel tuning. Specifically, KnowOS introduces three key innovations: **(1) Structured Knowledge Construction and Mapping**: We construct an OS-oriented Dual-layer Knowledge Graph (OD-KG) that maps high-level tuning objectives to corresponding low-level config options, ensuring a comprehensive and interpretable alignment. **(2) Knowledge-driven Configuration Generation**: We propose a systematic and effective kernel configuration generation strategy based on reasoning over the OD-KG, mitigating the hallucination issues typically encountered by LLMs. **(3) Continuous Knowledge Maintenance**: We design an efficient mechanism for continuously updating the OD-KG, allowing it to adapt incrementally to kernel updates without requiring complete model retraining.

We systematically evaluate the effectiveness of KnowOS in kernel tuning using two representative OS benchmarking suites: UnixBench Byte UnixBench Developers (1983) and LEBench Ren et al. (2019). Additionally, we assess its performance across four widely adopted and real-world applications: Nginx, Redis, Apache, and PostgreSQL. These applications span a diverse range of real-world workloads. KnowOS achieves **7.1× to 155.4×** speedups over baseline methods on synthetic benchmarks and up to **142%** performance improvement compared to default configurations on applications. These results demonstrate the effectiveness, efficiency, and scalability of KnowOS in automating kernel tuning, underscoring its practical value in real-world deployment scenarios.

## 2 PRELIMINARIES

**Definition 1: Kernel Space.** We model the Linux kernel space as a directed graph $\mathcal{S} = (O, E, C)$, where each node $o \in O$ represents a configurable option, and its value assignment $x$ is drawn from domain $D_o$. The edge set $E \subseteq O \times O$ encodes dependency relations between options: an edge $(o_i, o_j) \in E$ indicates that $o_j$ depends on $o_i$ and cannot be selected independently. The constraint function $C(x_i, x_j) \to \{\texttt{True}, \texttt{False}\}$ defines whether a pair of assignments is valid: $C(x_i, x_j) = \texttt{True}$ means that assigning $x_i$ to $o_i$ and $x_j$ to $o_j$ complies with kernel constraints.

**Definition 2: Kernel Configuration.** A kernel configuration $K = \{(o, x)\} \subseteq O \times D$ is a subset of options from the kernel space along with their assigned values, where each option $o$ is assigned a value $x$. The configuration $K$ is valid if all assigned values are within their domains, all dependencies in $E$ are respected, and all relevant constraints defined by $C$ are evaluated to $\texttt{True}$.

**Problem Formulation: Kernel Tuning Task.** Given a tuning objective $q$ and an evaluation function $P(K, q) \to \mathbb{R}$ that quantifies how well a configuration $K$ satisfies $q$, the kernel tuning task seeks a

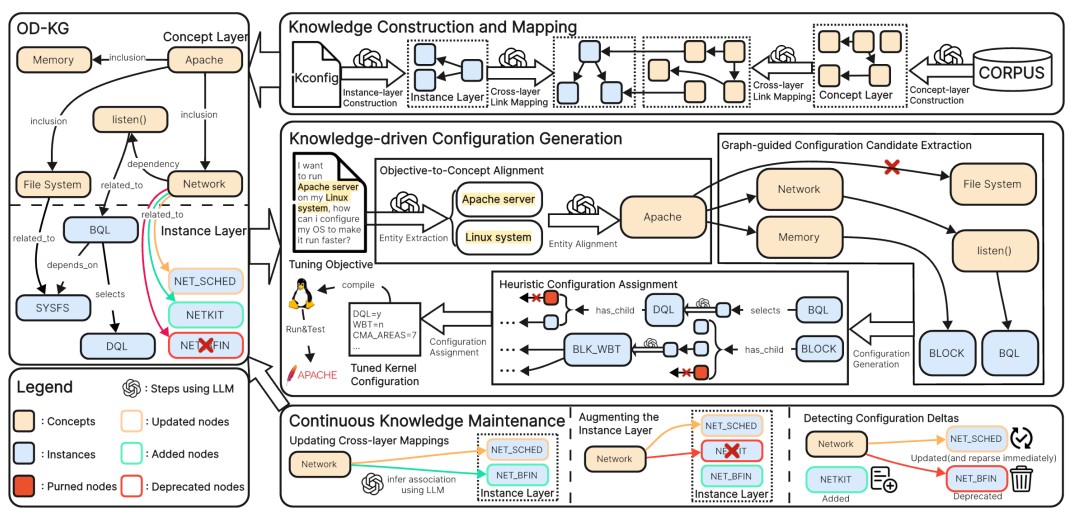

Figure 3: An overview of our KnowOS framework. First, we construct the OD-KG from a pre-built corpus (Knowledge Construction and Mapping 3.1). We then generate kernel configurations with the help of OD-KG (Knowledge-driven Configuration Generation 3.2). Since kernel config might be changed frequently, we need to add new configs, delete deprecated configs and update changed configs to OD-KG (Continuous Knowledge Maintenance 3.3)
.

valid configuration that maximizes $P(K, q)$ while satisfying all domain, dependency, and constraint requirements. Formally:

$$
\begin{aligned}
\text{Maximize} \quad & P(K, q), \quad K \subseteq O \times D \\
\text{Subject to} \quad & x_i \in D_{o_i} \quad \forall (o_i, x_i) \in K, \\
& \texttt{Dependencies}(K, E) = \texttt{True}, \\
& \texttt{Constraints}(K, C) = \texttt{True}
\end{aligned}
$$

## 3 METHOD: KNOWOS

In this section, we introduce KnowOS, as shown in Figure 3, which includes three major components: Structured Knowledge Construction and Mapping, Knowledge-driven Configuration Generation, and Continuous Knowledge Maintenance.

### 3.1 KNOWLEDGE CONSTRUCTION AND MAPPING

To bridge the semantic gap between high-level abstract tuning objectives and low-level concrete config options, inspired by Hao et al. (2021); Luo et al. (2023), KnowOS introduces a structured knowledge representation: an OS-oriented Dual-layer Knowledge Graph (OD-KG). This graph captures both domain concepts and kernel-specific configurations through three components:

- **Instance Layer** $\mathcal{G}_{\mathcal{I}} = (\mathcal{E}_{\mathcal{I}}, \mathcal{R}_{\mathcal{I}})$: Encodes kernel space $\mathcal{K}$, where entities $\mathcal{E}_{\mathcal{I}}$ denotes config options and relations $\mathcal{R}_{\mathcal{I}}$ represents dependencies and constraints extracted from the kernel space.

- **Concept Layer** $\mathcal{G}_{\mathcal{C}} = (\mathcal{E}_{\mathcal{C}}, \mathcal{R}_{\mathcal{C}})$: Captures domain knowledge, where $\mathcal{E}_{\mathcal{C}}$ represents generalized kernel tuning concepts and $\mathcal{R}_{\mathcal{C}}$ models their semantic relationships.

- **Cross-layer Links** $\mathcal{L} = \{(e_I, related\_to, e_C) \mid e_I \in \mathcal{E}_{\mathcal{I}}, e_C \in \mathcal{E}_{\mathcal{C}}\}$: Establishes semantic mappings from tuning concepts to their associated concrete config options.

The unified OD-KG is defined as $\mathcal{G} = (V, E)$, with vertex set $V = \mathcal{E}_{\mathcal{C}} \cup \mathcal{E}_{\mathcal{I}}$ and edge set $E = \mathcal{R}_{\mathcal{C}} \cup \mathcal{R}_{\mathcal{I}} \cup \mathcal{L}$. This structured representation enables interpretable and efficient reasoning from abstract tuning objectives to actionable configurations.

**Instance-layer Construction.** We construct configuration entities $\mathcal{E}_\mathcal{I}$ and the dependency relations $\mathcal{R}_\mathcal{I}$ among them by parsing the official Linux `Kconfig` file The Linux Foundation (2023). Dependency relations $\mathcal{R}_\mathcal{I}$ are identified using keyword-based extraction, covering four primary types:

$$\mathcal{R}_\mathcal{I} = \{(e_i, r, e_j) \mid r \in \{depends\_on, select, imply, has\_child\}, \ e_i, e_j \in \mathcal{E}_\mathcal{I}\} \tag{1}$$

Example: as illustrated in Figure 6, the extracted option "*config ZSWAP*" is encoded as entity *ZSWAP* $\in \mathcal{E}_\mathcal{I}$, while identified relations such as $(ZSWAP, depends\_on, SWAP)$ and $(ZSWAP, select, ZPOOL)$ are belong to $\mathcal{R}_\mathcal{I}$.

**Concept-layer Construction.** We extract the concept layer $\mathcal{G}_\mathcal{C}$ using few-shot in-context learning Brown et al. (2020) with a LLM. Prompts (Appendix A.1) are constructed from a curated corpus of OS kernel tuning materials, including benchmarks, research papers, official manuals, and domain datasets. The LLM first extracts tuning objectives as entities $\mathcal{E}_\mathcal{C}$, and then infers semantic relationships between pairs of entities to form the relation set $\mathcal{R}_\mathcal{C}$. Specifically, we define:

$$\mathcal{R}_\mathcal{C} = \{(e_i, r, e_j) \mid r \in \{inclusion, dependency, influence\}, \ e_i, e_j \in \mathcal{E}_\mathcal{C}\} \tag{2}$$

Example: Figure 6 shows the concept *I/O Reduction* and its inferred relationship: $(RAM\text{-}based\ Memory\ Pool, influence, I/O\ Reduction) \in \mathcal{R}_\mathcal{C}$.

**Cross-layer Link Mapping.** KnowOS uses LLMs to map config options $(\mathcal{E}_\mathcal{I})$ to relevant tuning objectives $(\mathcal{E}_\mathcal{C})$ based on their functional semantics. These links are expressed as:

$$\mathcal{L} = \{(e_I, \texttt{related\_to}, e_C) \mid e_I \in \mathcal{E}_\mathcal{I}, \ e_C \in \mathcal{E}_\mathcal{C}\} \tag{3}$$

Example: as shown in Figure 6, the link $(ZSWAP, \texttt{related\_to}, Swap\ Pages) \in \mathcal{L}$ captures the association between a low-level config option and a high-level memory tuning objective.

**Proposition 1.** *Dual-layer knowledge graph resolves semantic issues by concept-instance mappings. Proof.* We provide experimental results in Section 4.4 and theoretical proofs in Appendix B.1.

## 3.2 KNOWLEDGE-DRIVEN CONFIGURATION GENERATION

Unlike traversal-based methods She et al. (2013) that exhaustively search the kernel space, KnowOS uses a knowledge-driven approach over the OD-KG to identify relevant config options, reducing LLM hallucinations and search overhead by pruning irrelevant subspaces.

**Aligning Tuning Objectives with Kernel Concepts.** Given tuning objective $q$, KnowOS extracts textual entities $\mathcal{E}$ via semantic parsing (e.g., from $q =$ *"Optimize Linux for faster Apache server."*, we get $\mathcal{E} = \{\texttt{Linux}, \texttt{Apache}\}$). Each entity $e \in \mathcal{E}$ is mapped to a concept in $\mathcal{E}_\mathcal{C}$ through the mapping function $\phi : \mathcal{E} \to \mathcal{E}_\mathcal{C}$: if $e \in \mathcal{E}_C$, KnowOS identifies it through pattern matching $\psi_\text{PM}$; otherwise, LLM-based semantic matching $\psi_\text{LLM}$ is utilized to rephrase and match it to the most semantically similar concepts within $\mathcal{E}_C$:

$$\phi(e) = \begin{cases} \psi_\text{PM}(e) & \text{if } \psi_\text{PM}(e) \neq \emptyset \\ \psi_\text{LLM}(e) & \text{otherwise} \end{cases} \tag{4}$$

The aligned concept set $\mathcal{E}_\mathcal{C}{}^q = \bigcup_{e \in \mathcal{E}} \phi(e)$ captures the high-level semantics of $q$, grounding subsequent reasoning over the OD-KG.

**Graph-guided Relevant Configuration Extraction.** For each concept node $e_s \in \mathcal{E}_\mathcal{C}{}^q$, KnowOS employs LLM-based reasoning over the OD-KG to explore path $\pi(e_s) = \langle e_s \xrightarrow{r_1} e_1 \xrightarrow{r_2} \cdots \xrightarrow{r_n} e_n \rangle$ that may lead to relevant config options. Each path is assigned a relevance score $\rho(\pi(e_s))$:

$$\rho(\pi(e_s)) = \prod_{i=1}^{n} \sigma(r_i) \cdot \omega(e_i), \quad \text{where} \quad e_i \in V, \ r_i \in E \tag{5}$$

where $\sigma(r_i)$ denotes the semantic strength of relation $r_i$, and $\omega(e_i)$ reflects the contextual importance of node $e_i$, both estimated by the LLM. Config options with high relevance score (i.e., $\rho \geq \tau$) are aggregated into the candidate set $K_q$, where $\tau$ is a threshold to assess path reasonability:

$$K_q = \{e_i \in \mathcal{E}_\mathcal{I} \mid e_i \in \pi(e_s), \ e_s \in \mathcal{E}_\mathcal{C}{}^q, \ \rho(\pi(e_s)) \geq \tau\} \tag{6}$$

This step effectively filters the kernel space to those options most relevant to the tuning objective.

**Heuristic Inference for Option Value Assignment.** Given the candidate configuration set $K_q$ from the OD-KG, KnowOS constructs a valid configuration $K = \{(o, x)\} \subseteq O \times D$ that maximizes the tuning score $P(K, q)$, as defined in Section 2. KnowOS iteratively assigns values to each option $o_t \in K_q$ through LLM-based inference, guided by domain knowledge and structural constraints (prompts in Appendix A.2). At each step $t$, the system maintains a partial configuration $K_t$, selecting an unassigned option $o_t \in K_q \setminus \{o_{t-1} \mid (o_{t-1}, \cdot) \in K_t\}$. The value assignment is inferred as:

$$x_t = \texttt{LLM\_Infer}(o_t \mid \mathcal{E}_C^q, \mathcal{G}, K_t), \tag{7}$$

where $\mathcal{E}_C^q$ denotes tuning concepts aligned with $q$, $\mathcal{G}$ is the relevant OD-KG subgraph, and $K_t$ provides current context. A valid inferred value $x_t$ must satisfy: (1) $x_t \in D_{o_t}$, (2) $\texttt{Dependencies}(K_t \cup \{(o_t, x_t)\}, E) = \texttt{True}$, and (3) $\texttt{Constraints}(K_t \cup \{(o_t, x_t)\}, C) = \texttt{True}$.

**Performance-aware Final Configuration Generation.** To further enhance configuration quality, KnowOS optionally selects the assignment $x_t^* \in D_{o_t}$ that maximizes the estimated tuning score $P(K, q)$ among all valid candidates. The configuration is then updated as: $K_{t+1} = K_t \cup \{(o_t, x_t^*)\}$. This process repeats until all options in $K_q$ have been assigned. The final configuration $K_T$ is both valid and semantically aligned with the tuning objective $q$, while aiming to maximize the performance metric $P(K, q)$. The complete assignment procedure is detailed in Algorithm 1.

$$K_T = \bigcup_{t=0}^{T-1} \left\{ (o_t, x_t^*) \,\middle|\, x_t^* = \arg\max_{x \in D_{o_t}} \left\{ P(K_t \cup \{(o_t, x)\}, q) \mid \texttt{Valid}(K_t \cup \{(o_t, x)\}) = \texttt{True} \right\} \right\}. \tag{8}$$

**Proposition 2.** *Knowledge-driven reasoning over KG mitigates LLM hallucinations in kernel tuning.*
*Proof.* We provide experimental results in Section 4.5 and theoretical proofs in Appendix B.2.

### 3.3 CONTINUOUS KNOWLEDGE MAINTENANCE

As the Linux kernel evolves rapidly with frequent releases and feature updates, maintaining an up-to-date and accurate knowledge graph becomes critical. To this end, KnowOS adopts an incremental update strategy to continuously evolve the OD-KG with minimal overhead.

Let $S^{(t)} = (O^{(t)}, E^{(t)}, C^{(t)})$ denote the kernel space at version $t$ as defined in Section 2, and $S^{(t+1)}$ for the next version. The goal is to update the instance layer $\mathcal{G}_{\mathcal{I}}^{(t)} = (\mathcal{E}_{\mathcal{I}}^{(t)}, \mathcal{R}_{\mathcal{I}}^{(t)})$ and cross-layer links $\mathcal{L}^{(t)}$ to reflect $S^{(t+1)}$, while preserving existing concept-level semantics $\mathcal{G}_{\mathcal{C}}$.

**Step 1: Detecting Configuration Deltas.** We begin by computing the configuration delta between two consecutive kernel versions to capture changes in the option set. Specifically, we identify newly added options as $\Delta O_{\text{add}} = o \in O^{(t+1)} \mid o \notin O^{(t)}$ and deprecated options as $\Delta O_{\text{del}} = o \in O^{(t)} \mid o \notin O^{(t+1)}$. Additionally, for options that persist across versions but exhibit changes in their domain definitions or dependency relations, we re-parse and update their corresponding entities and edges to reflect the latest semantics.

**Step 2: Augmenting the Instance Layer.** For each new option $o \in \Delta\mathcal{O}_{\text{add}}$, we add its corresponding new entity $e_o$ to $\mathcal{E}_{\mathcal{I}}^{(t+1)}$, then extract its relations $\mathcal{R}_o$ and insert them into $\mathcal{R}_{\mathcal{I}}^{(t+1)}$:

$$\mathcal{R}_o = \{(e_o, r, e_{o'}) \mid r \in \{\texttt{depends\_on}, \texttt{select}, \texttt{imply}, \texttt{has\_child}\}, e_{o'} \in \mathcal{O}^{(t+1)}\} \tag{9}$$

For each deprecated option $o \in \Delta\mathcal{O}_{\text{del}}$, we delete the associated entity $e_o$ and all related relations:

$$\mathcal{E}_{\mathcal{I}}^{(t+1)} \leftarrow \mathcal{E}_{\mathcal{I}}^{(t)} \setminus \{e_o \mid o \in \Delta\mathcal{O}_{\text{del}}\} \tag{10}$$

$$\mathcal{R}_{\mathcal{I}}^{(t+1)} \leftarrow \mathcal{R}_{\mathcal{I}}^{(t)} \setminus \{(e_{o_i}, r, e_{o_j}) \mid e_{o_i} = e_o \vee e_{o_j} = e_o\} \tag{11}$$

**Step 3: Updating Cross-layer Mappings.** For each new instance entity $e_I \in \mathcal{E}_{\mathcal{I}}^{(t+1)} \setminus \mathcal{E}_{\mathcal{I}}^{(t)}$, we invoke a LLM to infer its semantic association with concept-layer entities $e_C \in \mathcal{E}_{\mathcal{C}}$, forming new cross-layer links:

$$\Delta\mathcal{L}^{(t+1)} = \{(e_I, \texttt{related\_to}, e_C)\} \tag{12}$$

These links ensure that newly introduced kernel options remain interpretable through high-level domain knowledge mapping. Deprecated options have their cross-layer links removed accordingly.

**Proposition 3.** *Continuous knowledge maintenance ensures tuning accuracy and robustness.*
*Proof.* We provide experimental results in Section 4.6 and theoretical proofs in Appendix B.3.

# 4    EXPERIMENTS

We evaluate the effectiveness and generalizability of KnowOS through a series of empirical studies designed to answer the following research questions (RQs): **RQ1:** How does KnowOS compare with existing baselines in kernel tuning? **RQ2:** Does the main component of KnowOS work? **RQ3:** Does KnowOS effectively address the knowledge mapping challenge? **RQ4:** To what extent does KnowOS mitigate LLM hallucinations in kernel tuning? **RQ5:** Can KnowOS remain effective across evolving kernel versions? **RQ6:** How does KnowOS perform in real-world application scenarios?

## 4.1    EXPERIMENTAL SETUP

**Linux Distributions.** To cover diverse usage scenarios of the Linux system, including Desktop, Server, IoT, Cloud, Embedded system, etc., we evaluate four widely adopted Linux distributions: **Ubuntu**, **Fedora**, **Debian**, and **openEuler**. Detailed specifications are provided in Table 4.

**Benchmarks.** Kernel performance is assessed using two benchmarking suites: (1) **UnixBench** Byte UnixBench Developers (1983), a macro-benchmark aggregating sub-tests scores (e.g., context switching, pipe throughput) to measure overall system performance, and (2) **LEBench** Ren et al. (2019), a micro-benchmark for fine-grained kernel operations (e.g., fork, mmap, pagefault).

**Applications.** We assess real-world impact using four representative applications: Nginx Web Server Nginx, Inc. (2004), Redis Key-Value Store Salvatore Sanfilippo (2009), Apache HTTP Server Apache Software Foundation (1995), and PostgreSQL Database Group (1996). Redis is evaluated using Redis Benchmark Sanfilippo (2009), Apache and Nginx via ApacheBench Apache Software Foundation (1997), and PostgreSQL with sysbench Akopytov (2004).

**Hardware.** Our experiments span a variety of hardware platforms: three laptops (Intel i7-1165G7, i5-11400H, i7-13700) and a Xeon E5-2680 v4 workstation for distribution-level tests, and an AMD Ryzen 9 7950X system for application-level benchmarks.

**Baselines.** We compare KnowOS against three baselines: (1) **Default Configuration**: hand-tuned by experts, (2) **Vanilla LLM**: a direct use of LLM to generate configurations with bootability checks, and (3) **AutoOS** Chen et al. (2024): a LLM-based framework using a state machine for kernel tuning.

**Implementations.** All methods use **GPT-4o-mini** under consistent runtime settings. Each experiment is repeated for at least 15 independent runs per target, with the best-scoring configuration retained. For fairness, AutoOS uses its best publicly released configurations, reflecting peak performance.

Table 1: Best UnixBench results across four distributions (higher values indicate better performance). The bolded values represent the best score for each distribution. Abbreviations: ET = Execl Throughput, FC = File Copy, PT = Pipe Throughput, CS = Context Switching, PC = Process Creation, SS = Shell Scripts, SC = System Call.

| | Dhrystone | Whetstone | ET | FC 1024 | FC 256 | FC 4096 | PT | CS | PC | SS 1 | SS 8 | SC | Total Score |
|---|---|---|---|---|---|---|---|---|---|---|---|---|---|
| | | | | | | **Ubuntu** | | | | | | | |
| Default | 5182 | 1842 | 1489 | 5466 | 3863 | 9629 | 2866 | 864 | 1145 | 4205 | 9003 | 2529 | 3099 |
| LLM | 5538 | 1863 | 1495 | 5840 | 3627 | 9591 | 2770 | 859 | 1150 | 4200 | 8987 | 2413 | 3098 (-0.0%) |
| AutoOS | **5616** | **1864** | 1533 | 5976 | 3819 | 9458 | 2945 | 854 | 1150 | 4241 | 9032 | 2527 | 3154 (+1.8%) |
| KnowOS | 5525 | 1848 | **1628** | **6266** | **4105** | **10079** | **3091** | **897** | **1231** | **4587** | **9816** | **2684** | **3318 (+7.1%)** |
| | | | | | | **Fedora** | | | | | | | |
| Default | **5408** | **1815** | 278 | 1284 | 778 | 2464 | 456 | 150 | 366 | 1095 | 4186 | 175 | 846 |
| LLM | 5394 | 1810 | **286** | 1316 | 843 | 2871 | 480 | 173 | 392 | **1182** | 4530 | 187 | 902 (+6.6%) |
| AutoOS | 4969 | 1669 | 281 | 1302 | 833 | 2613 | 458 | 147 | 397 | 1078 | 3981 | 177 | 846 (+0.0%) |
| KnowOS | 4870 | 1688 | 258 | **1319** | **922** | **2885** | **558** | **239** | **400** | 1155 | **4542** | **217** | **936 (+10.6%)** |
| | | | | | | **Debian** | | | | | | | |
| Default | 6271 | **2044** | 1315 | 5031 | 3162 | 10029 | 2300 | 276 | 1199 | 4689 | **10702** | 1604 | 2721 |
| LLM | 6066 | 2002 | 1131 | 5584 | 3569 | 10473 | 2572 | 344 | 1045 | 4278 | 9671 | 1959 | 2782 (+2.2%) |
| AutoOS | **6346** | 2041 | **1356** | 6646 | 4143 | 12070 | 2964 | 405 | **1209** | **4715** | 10695 | **2404** | 3169 (+16.5%) |
| KnowOS | 6298 | 2035 | 1221 | **7538** | **4896** | **13828** | **3522** | **514** | 1098 | 4531 | 10385 | 2273 | **3305 (+21.4%)** |
| | | | | | | **OpenEuler** | | | | | | | |
| Default | 3442 | 1300 | 210 | 614 | 372 | 1565 | 240 | 42 | 88 | 441 | 3650 | 123 | 442 |
| LLM | 3470 | 1291 | 332 | 603 | 365 | 1530 | 227 | 41 | 74 | 380 | 2585 | 121 | 430 (-2.7%) |
| AutoOS | 3164 | 1200 | 237 | 2960 | 1989 | 6302 | 1393 | 40 | 107 | 603 | 3955 | 1071 | 945 (+113.7%) |
| KnowOS | **3500** | **1315** | 251 | **3674** | **2405** | **7323** | **1635** | **54** | **135** | **648** | **4256** | **1643** | **1129 (+155.4%)** |

## 4.2 OVERALL KERNEL PERFORMANCE (RQ1)

**KnowOS Performance Advantages.** Table 1 summarizes UnixBench results across four distributions. KnowOS consistently outperforms all baselines, with relative improvements of **7.1%**, **10.6%**, **21.4%**, and **155.4%**, demonstrating its effectiveness in kernel tuning. These gains are attributed to KnowOS's structured knowledge mapping and knowledge-driven configuration generation strategy, which effectively aligns tuning goals with config options.

**Limitations of Vanilla LLM and AutoOS.** In contrast, **Vanilla LLM** shows inconsistent performance, with some configurations even reducing throughput (e.g., -2.7% on openEuler). The lack of interaction with kernel space limits its ability to optimize performance effectively. **AutoOS**, while delivering notable improvements in specific cases (e.g., +16.5% on Debian), is constrained by its limited kernel knowledge, preventing full exploitation of relevant config options.

**Trade-offs in Sub-test and Holistic Optimization.** Interestingly, the trade-offs in sub-test performance (e.g., AutoOS leads in Dhrystone/Whetstone for Ubuntu, while KnowOS excels in other sub-tests) highlight KnowOS's holistic optimization strategy, which prioritizes overall system performance rather than isolated improvements, mitigating conflicts in concurrent parameter optimizations.

## 4.3 ABLATION STUDY (RQ2)

**Experiment Objective.** To evaluate the contributions of the key components in KnowOS, we conduct an ablation study on Ubuntu using UnixBench (Table 2).

Table 2: Ablation study of KnowOS on Ubuntu using UnixBench. **Default** denotes the system's default configuration, **w/o KG** removes the OD-KG knowledge base, and **w/o Mapping** removes the structured knowledge mapping strategy. Sub-test abbreviations follow Table 1.

| | Dhrystone | Whetstone | ET | FC 1024 | FC 256 | FC 4096 | PT | CS | PC | SS 1 | SS 8 | SC | Total Score |
|---|---|---|---|---|---|---|---|---|---|---|---|---|---|
| Default | 5182 | 1842 | 1489 | 5466 | 3863 | 9629 | 2866 | 864 | 1145 | 4205 | 9003 | 2529 | 3099 |
| w/o Mapping | 5495 | 1818 | 1504 | 5710 | 3564 | 9564 | 2587 | 802 | 1159 | 4069 | 8705 | 2219 | 3010 (-2.9%) |
| w/o KG | 5389 | 1826 | 1530 | 5879 | 3781 | 9596 | 2843 | 862 | 1172 | 4277 | 8923 | 2373 | 3115 (+0.5%) |
| KnowOS | **5525** | **1848** | **1628** | **6266** | **4105** | **10079** | **3091** | **897** | **1231** | **4587** | **9816** | **2684** | **3318 (+7.1%)** |

**Effect of Knowledge Mapping.** Disabling the knowledge mapping module (*w/o Mapping*) significantly degrades performance. Without mapping tuning objectives to config options in OD-KG, the LLM generates less coherent and misaligned reasoning paths, emphasizing the importance of knowledge mapping in grounding LLM behavior in domain-specific tasks.

**Effect of OD-KG.** Removing the OD-KG (*w/o KG*) knowledge base leads to only marginal improvements, indicating that while the LLM can extract basic intent from user input, the lack of structured domain knowledge limits its decision-making. This highlights the knowledge base's key role in enhancing KnowOS's informed, knowledge-driven decisions.

## 4.4 STRUCTURED KNOWLEDGE MAPPING ENHANCES FINE-GRAINED TUNING (RQ3)

**Experiment Objective.** We assess KnowOS's ability to map high-level tuning objectives to low-level config options by benchmarking individual system calls using LEBench. In this experimental setup, tuning objectives are defined at the system call level, allowing us to observe the impact of each method on fine-grained kernel operations.

**Result Analysis** Figure 4 shows the relative latency change across various kernel operations. KnowOS consistently reduces latency across most operations, including `fork`, `thr-create`, `mmap`, `page-fault`, and `epoll`, demonstrating effective knowledge-driven mapping from abstract tuning objectives to concrete config options. In contrast, both AutoOS and vanilla LLM yield mixed results, improving some operations while degrading others.

**Summary.** These findings confirm that structured knowledge mapping employed in KnowOS enables precise, goal-aligned tuning at the system call level. This approach effectively addresses a key challenge in kernel configuration: translating high-level, abstract tuning objectives into low-level, concrete config options that optimize performance.

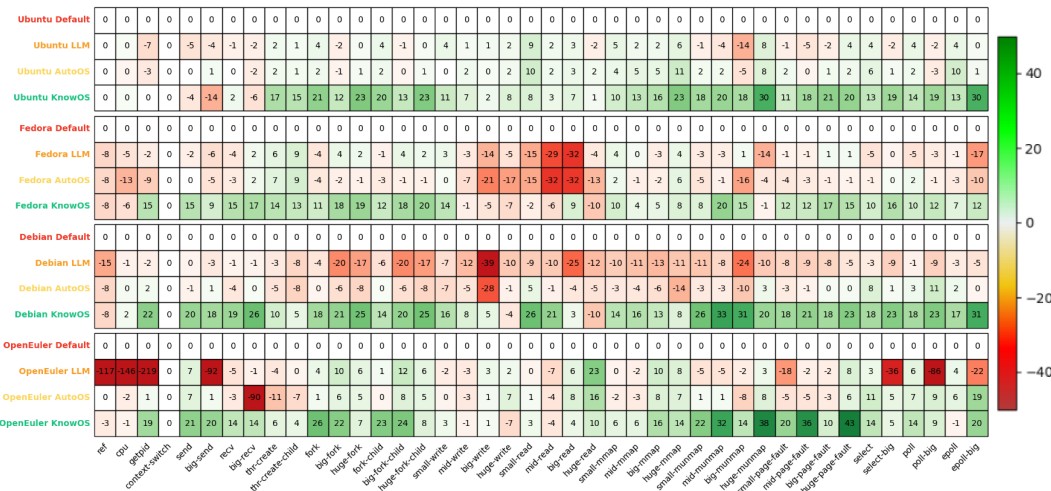

Figure 4: Result of LEBench: the heatmap shows the relative latency changes in kernel operations for each method, compared to the default configuration. Green indicates reduced latency (better), while red denotes increased latency (worse).

## 4.5 MITIGATING LLM HALLUCINATIONS (RQ4)

Table 3: LLM hallucination test across 8 tuning objectives. Results include compile errors (CE), boot errors (BE), average UnixBench scores (Score), score variance, and the best score among valid runs.

|  | cfg-1 | cfg-2 | cfg-3 | cfg-4 | cfg-5 | cfg-6 | cfg-7 | cfg-8 | Variance($\sigma^2$) | Best |
|---|---|---|---|---|---|---|---|---|---|---|
| AutoOS | 2776 | 1779 | CE | BE | 2660 | CE | 1187 | CE | 426796 | 2776 |
| KnowOS | 2825 | 2813 | 2883 | BE | 2861 | 2664 | BE | 2817 | 5936 | 2861 |

**Experiment Objective.** To evaluate KnowOS's robustness against LLM hallucinations, we assess the validity of configurations across eight tuning objectives, comparing configurations generated by KnowOS and vanilla LLM. Each configuration is labeled as: (1) **CE** for compilation error; (2) **BE** for boot error; and (3) **Score** for valid UnixBench score. Results are summarized in Table 3.

**Result Analysis.** KnowOS compiled all configurations with only two boot errors, showing low variance (5936) and achieving a best score of 2861. In contrast, vanilla LLM produced three configurations with compile errors, one with a boot error, and four that ran UnixBench, exhibiting high variance (426796) and a best score of 2776.

**Summary.** These results demonstrate that KnowOS significantly improves configuration validity and stability, highlighting the effectiveness of its knowledge-driven generation process in mitigating the hallucinations that are common in vanilla LLM-based approaches.

## 4.6 ADAPTABILITY ACROSS KERNEL VERSIONS (RQ5)

**Experiment Objective.** Linux kernels evolve rapidly, with config options frequently changed across versions. To evaluate KnowOS's adaptability, we assess its performance on four Ubuntu releases (14.04, 16.04, 18.04, 20.04) with kernel versions 3.13, 4.15, 5.4, and 6.2, using UnixBench.

**Result Analysis.** As shown in Figure 5(a), **KnowOS consistently delivers performance gains across versions**, even with significant changes. This highlights KnowOS's robustness in adapting to evolving kernel versions with minimal retraining.

**Summary.** This adaptability stems from OD-KG's continuous knowledge maintenance, which tracks upstream differences and incrementally refines the knowledge graph. These results confirm KnowOS's viability as a long-term solution for kernel tuning in dynamically evolving environments.

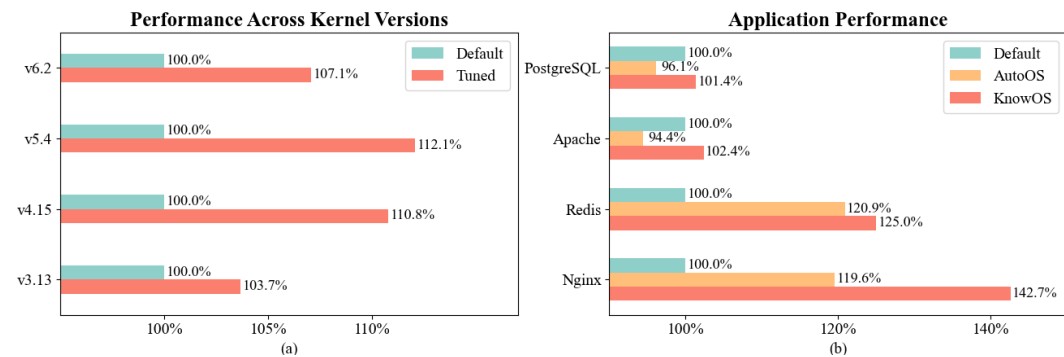

Figure 5: Performance evaluation across kernel versions (a) and real-world applications (b). The x-axis shows the performance normalized to the default configuration. Nginx and Apache are measured in **Requests per Second**, Redis in **Operations per Second**, and PostgreSQL in **Queries per Second**.

### 4.7 REAL-WORLD APPLICATION EVALUATION (RQ6)

**Experiment Objective.** To evaluate the practical utility of KnowOS, we conduct a comprehensive study on four widely deployed real-world applications: Nginx, Apache, Redis, and PostgreSQL. These applications represent diverse workloads, including CPU-bound, I/O-bound, and memory-intensive.

**Result Analysis.** Figure 5(b) summarizes the performance gains—throughput and latency—achieved by KnowOS compared to the default configuration. KnowOS consistently delivers significant gains, with Redis throughput increasing by up to 25.0% and Nginx latency reduced by up to 42.7%.

**Summary.** These results demonstrate KnowOS's effectiveness in identifying configuration optimizations that are difficult to achieve through traditional methods. The findings validate KnowOS as a robust and adaptable solution for kernel tuning in real-world environments.

## 5 RELATED WORK

**OS Kernel Tuning.** Previous kernel optimization efforts include both manual and automated methods. Network-specific tuning is studied in Evang & Dreibholz (2024); Schwarz et al. (2024), while Kroth et al. (2024); Martin et al. (2021) optimizes kernel performance via machine learning. LEBench Ren et al. (2019) traces performance regressions to specific config changes, and DeepPerf Ha & Zhang (2019) predicts performance using sparse deep neural networks. Kernel debloating is addressed in Kuo et al. (2022), and configuration conflict resolution is explored by Franz et al. (2021). AutoOS Chen et al. (2024) combines LLMs with state-machine optimization for AIoT-specific tuning.

**Knowledge-Driven LLMs in Software Engineering.** LLMs are increasingly applied to software engineering tasks. Fine-tuning methods for code generation are discussed in Jiang et al. (2024); Weyssow et al. (2023); Liu et al. (2024), while vulnerability discovery is addressed in Ghosh et al. (2025). A system for LLM-based code synthesis requiring deep reasoning is proposed in Li et al. (2022). LLMs for incident mitigation are evaluated in Ahmed et al. (2023), and prompt-based config validation is explored in Lian et al. (2024). An agentless repair pipeline for software bugs is introduced in Xia et al. (2024).

## 6 CONCLUSION

We introduce **KnowOS**, a knowledge-driven framework leveraging LLMs for OS kernel tuning. By integrating a dual-layer, OS-specific knowledge graph with targeted retrieval, KnowOS bridges abstract tuning goals and concrete config options. Extensive evaluations demonstrate its superiority in generating performant, stable, and adaptable kernel configurations. This work highlights the potential of structured knowledge integration in enhancing LLM-based system optimization.

## ETHICS STATEMENT

KnowOS leverages knowledge-driven LLMs to automate Linux kernel tuning, offering efficiency gains but also raising ethical concerns. While it reduces reliance on manual tuning, improper configurations may destabilize system performance, requiring strict validation. Continuous knowledge maintenance helps mitigate LLM hallucinations, but accuracy and reliability must be ensured. Responsible use, including transparency and ongoing validation, is essential for ethical deployment.

## REPRODUCIBILITY

To ensure the reproducibility of KnowOS, the source code is publicly available at the following URL: https://anonymous.4open.science/r/KnowOS-B274. These measures are intended to facilitate the verification and replication of our results by other researchers in the field.

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

# APPENDIX

# A PROMPTS USED IN KNOWOS

## A.1 OD-KG CONSTRUCTION PROMPTS

**Entity Extraction & Relation Identification.**    The first step in constructing the knowledge graph from structured Kconfig data involves the use of `Kconfiglib`. Each config option within the kernel is represented as an entity. Dependencies such as "depends on" or "select" are modeled as directed relations between entities, ensuring the accurate representation of kernel space dependencies. For textual data, such as the help text describing a config option, the description is encapsulated in the format: "Config `xxx` description: text". If no description is available, the prompt directly uses the config option's name as context.

For each of these configurations, we generate specific prompts that guide LLMs to detect the entities and identify their relationships. This procedure is illustrated in Figure 6 and exemplified in the prompt shown in Figure 7. These prompts ensure the model accurately extracts the essential features and relationships from both structured and unstructured data sources.

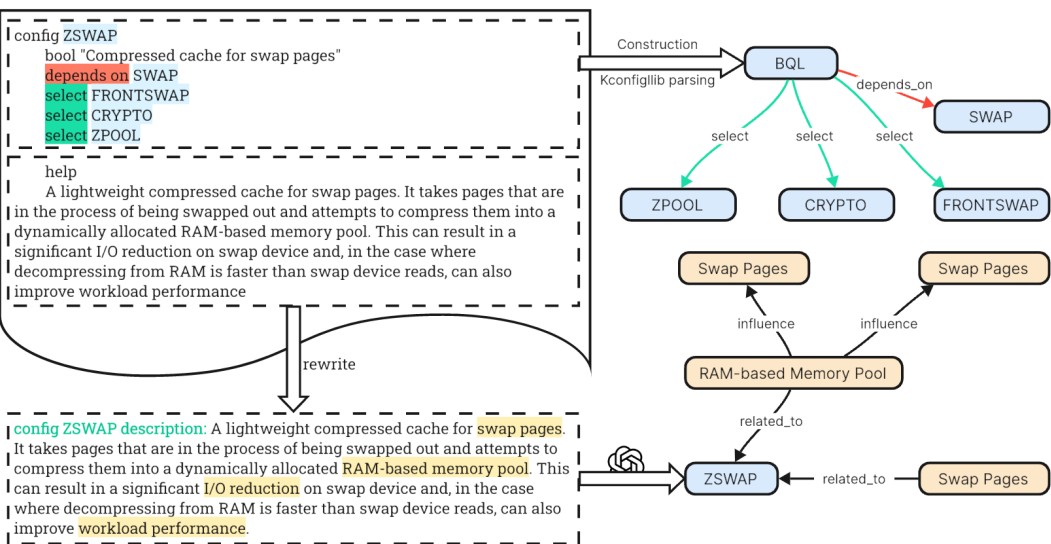

Figure 6: Entity & Relation Extraction Process

## A.2 LARGE LANGUAGE MODELS EXPLORE KERNEL SPACE PROMPTS

The Linux kernel space consists of several configuration types: `Bool`, `Choice`, `Menu`, and `Value`. Each configuration type is addressed with tailored interaction templates designed for LLMs. These prompts are intended to guide the LLM in exploring and selecting the most relevant kernel configurations that align with a given tuning objective.

### A.2.1 BOOL

The `Bool` type configuration has two possible values: `on` and `off`. For efficient exploration, we group these configurations into sets of up to nine, prompting the LLM to evaluate their collective impact on the target optimization objective. Instead of directly querying for the configuration values, the LLM is asked to indicate the effect of each configuration group on the target. The possible responses are: "increase" (positive effect), "decrease" (negative effect), and "cannot determine" (neutral or indeterminate effect). This method reduces computational overhead and focuses the model's attention on the relative impact of configuration sets. The corresponding prompt is shown in Figure 8.

Given a text document that describes either a linux kernel configuration or a computer science knowledge text, identify all entities from the text and all relationships among the identified entities. You can follow the steps below:
1. Identify all the entities with the following information:
- entity_name: Name of the entity, and capitalized the name.
- entity_description: Comprehensive description of the entity's attributes and activities
Format each entity as ("entity"&<entity_name>&<entity_description>)
2. From the entities identified in step 1, identify all pairs of (source_entity, target_entity) that are *clearly related* to each other.
For each pair of related entities, extract the following information:
- source_entity: name of the source entity, as identified in step 1
- target_entity: name of the target entity, as identified in step 1
Format each relationship as ("relationship"&<source_entity>&<target_entity>)
3. Return output as a single list of all the entities and relationships identified in steps 1 and 2.
Here are some examples:
Example 1:
...
############################
Example 2:
...
############################
Here's the real input:
...
Output:

Figure 7: Entity & Relation Extraction Prompt

KNOWLEDGE = {}    TARGET = {}    CONFIGS = {}
Q: "I want to explore the config options related to TARGET in the Linux kernel configurations. Please choose the configs concerned with TARGET in the CONFIGS as much as possible. For each concerned config related to TARGET, you should determine whether it will increase or decrease TARGET. If it increases TARGET, output [CONFIG increase]. If it decrease TARGET, output [CONFIG decrease]. If a config is not related to TARGET, output [CONFIG - cannot determine impact without specific context]. You can reference the knowledge in KNOWLEDGE. I also have to gurantee the success boot of the OS after selecting. Answer in the following form, without any explanation, just answer in pure text form, give me config names in my given CONFIGS, each line represents a single config like this:
[config_name_1 increase]
[config_name_2 decrease]
...
[config_name_n increase]

Figure 8: Prompt for `Bool` type config options

The terms "KNOWLEDGE", "TARGET", and "CONFIGS" are dynamically replaced with relevant values during the prompt generation. Knowledge is gathered by querying LightRAG, and the nine configurations are split into three groups, each of which is queried individually. The results from each group are concatenated to form the comprehensive knowledge used for decision-making, as illustrated in Figure 9.

Q:
KNOWLEDGE =
Impact on Unixbench Total Score: Unixbench is a popular benchmarking tool used to measue...
 1. EFI (Extensible Firmware Interface): EFI is related to firmware-level configuration and ...
 2. KEXEC (Kernel Execute): KEXEC allows a system to boot into a new kernel without going ...
 3. RANDOMIZE_MEMORY: `RANDOMIZE_MEMORY` involves randomizing the layout of ...
...
TARGET = the unixbench total score
CONFIGS =
EFI runtime service support (EFI)
kexec system call (KEXEC)
Randomize the kernel memory sections (RANDOMIZE_MEMORY)
...
I want to explore the config options related to TARGET in the Linux kernel configurations. Please choose the configs concerned with TARGET in the CONFIGS as much as possible. For each concerned config related to TARGET, you should determine whether it will increase or decrease TARGET. If it increases TARGET, output [CONFIG increase]. If it decrease TARGET, output [CONFIG - cannot determine impact without specific context]. You can reference the knowledge in KNOWLEDGE. I also have to gurantee the success boot of the OS after selecting. Answer in the following form, without any explanation, just answer in pure text form, give me config names in my given CONFIGS, each line represents a single config like this:
[config_name_1 increase]
[config_name_2 decrease]
...
[config_name_n increase]

A:   [EFI increase]
     [KEXEC - cannot determine impact without specific context]
     [RANDOMIZE_MEMORY decrease]
     ...

Figure 9: Instance of a `Bool` type config option query

### A.2.2 CHOICE

A `Choice` configuration contains multiple sub-configurations, from which only one can be selected at any given time. To efficiently manage this, LLMs are prompted to choose the most appropriate configuration based on its relevance to the target objective. The prompt for this configuration type is depicted in Figure 10.

KNOWLEDGE = {}     TARGET = {}     CONFIGS = {}
Q: I want to explore the config options related to TARGET in the Linux kernel configurations. The CONFIGS I gave you are choices of a config, and you need to choose which config is most likely related to TARGET. Give me only one config in my give CONFIGS. You can reference the knowledge in KNOWLEDGE. I also have to gurantee the success boot of the OS after selecting. Answer in the following form, without any explanation, just answer in pure text form, each line represents a single config like this:
[config_name]

Figure 10: Prompt for `Choice` type config options

Just like with `Bool` types, the terms "KNOWLEDGE", "TARGET", and "CONFIGS" are used in the prompt and replaced with specific values. An example query is shown in Figure 11, demonstrating how the LLM is tasked with selecting the most suitable configuration.

Q:
KNOWLEDGE =
Analyzing how configuration setting might impact the "unixbench total score" involves ...
  1. X86_INTEL_TSX_MODE_OFF: This configuration option disables Intel's Transactional ...
  2. X86_INTEL_TSX_MODE_ON: Enabling this setting turns on TSX on compatible hardware ...
  3. X86_INTEL_TSX_MODE_AUTO: This option results in TSX being enabled on hardware ...
TARGET = the unixbench total score
CONFIGS =
off (X86_INTEL_TSX_MODE_OFF)
on (X86_INTEL_TSX_MODE_ON)
auto (X86_INTEL_TSX_MODE_AUTO)
I want to explore the config options related to TARGET in the Linux kernel configurations. The CONFIGS I gave you are choices of a config, and you need to choose which config is most likely related to TARGET. Give me only one config in my give CONFIGS. You can reference the knowledge in KNOWLEDGE. I also have to gurantee the success boot of the OS after selecting. Answer in the following form, without any explanation, just answer in pure text form, each line represents a single config like this:
[config_name]

A:  X86_INTEL_TSX_MODE_AUTO

Figure 11: Instance of a `Choice` type config option query

### A.2.3 MENU

A `Menu` configuration type includes multiple sub-configurations, and the task is to determine which of these are relevant to the optimization target. The prompt for this type is designed to guide the LLM to identify the relevant configurations based on their relation to the target. This process is illustrated in Figure 12.

KNOWLEDGE = {}     TARGET = {}     CONFIGS = {}
Q: I want to explore the config options related to TARGET in the Linux kernel configurations. Please choose the directories concerned with TARGET in the CONFIGS as much as possible. You can reference the knowledge in KNOWLEDGE. I also have to gurantee the success boot of the OS after selecting. Answer in the following form, without any explanation, just answer in pure text form, give me config names in my given CONFIGS, each line represents a single config like this:
[directory_name_1]
...
[directory_name_n]

Figure 12: Prompt for `Menu` type config options

As with previous configuration types, the "KNOWLEDGE", "TARGET", and "DIRECTORIES" are dynamically replaced based on the specific query context. An instance of such a query is shown in Figure 13, which illustrates how the menu-related options are identified and evaluated.

### A.2.4 VALUE

The `Value` configuration type includes various types, such as integers, hexadecimal values, and strings. The prompt for this configuration type is shown in Figure 14. The process for handling the "KNOWLEDGE", "TARGET", and "CONFIGS" remains consistent with previous types.

Q:
KNOWLEDGE =
The UnixBench total score is a benchmark suite designed to test the performance of Unix-like operating systems ...
  1. General Setup: This category often includes fundamental settings that control ...
  2. Processor Type and Features: This category is crucial as it allows for adjustments ...
  3. General Architecture-Dependent Options: This configuration category appears to include ...
...
TARGET = the unixbench total score
CONFIGS =
0 General setup
1 Processor type and features
2 General architecture-dependent options
...
I want to explore the config options related to TARGET in the Linux kernel configurations. Please choose the directories concerned with TARGET in the CONFIGS as much as possible. You can reference the knowledge in KNOWLEDGE. I also have to gurantee the success boot of the OS after selecting. Answer in the following form, without any explanation, just answer in pure text form, give me config names in my given CONFIGS, each line represents a single config like this:
[directory_name_1]
...
[directory_name_n]

A:   1 Processor type and features
      7 Memory Management options
      8 Device Drivers
      ...

Figure 13: Instance of a `Menu` type config option query

KNOWLEDGE = {}      TARGET = {}      CONFIGS = {}
I'm looking for the Linux kernel's menuconfig options that could potentially affect TARGET. I have listed some numeric config options listed in menuconfig, along with their corresponding value ranges. For each option, please select a value that may help improve TARGET. If the option is not related to TARGET reset it to the default value. For instance, if you are given: 'maximum CPU number (1=>2 2=>4) (cpunum) (1)', your response should be: 'maximun CPU number (1=>2 2=>4) (cpunum) (2)' because when the CPU number is more, the speed is usually better.
Config input format: [option name] (default value)
Value output format: [option name] (recommended value)
Attention! Please provide your recommended values without extra explanations or additional details. Only suggest options that could possibly help TARGET, and do not add units next to the numbers. You can reference the knowledge in KNOWLEDGE. Below are the numeric config options for your recommendations:
CONFIGS

Figure 14: Prompt for `Value` type config options

An example query for the `Value` type is shown in Figure 15, demonstrating how the LLM is tasked with selecting or generating values based on the knowledge and constraints provided.

Q:
KNOWLEDGE =
In the context of improving performance metrics such as the total score from Unixbench, ...
  1. PHYSICAL_START: The `PHYSICAL_START` configuration denotes the physical address ...
  2. PHYSICAL_ALIGN: Similarly, the `PHYSICAL_ALIGN` configuration refers to the alignment ...
TARGET = the unixbench total score
CONFIGS =
PHYSICAL_START (0x1000000)
PHYSICAL_ALIGN (0x200000)
I'm looking for the Linux kernel's menuconfig options that could potentially affect TARGET. I have listed some numeric config options listed in menuconfig, along with their corresponding value ranges. For each option, please select a value that may help improve TARGET. If the option is not related to TARGET reset it to the default value. For instance, if you are given: 'maximum CPU number (1=>2 2=>4) (cpunum) (1)', your response should be: 'maximun CPU number (1=>2 2=>4) (cpunum) (2)' because when the CPU number is more, the speed is usually better.
Config input format: [option name] (default value)
Value output format: [option name] (recommended value)
Attention! Please provide your recommended values without extra explanations or additional details. Only suggest options that could possibly help TARGET, and do not add units next to the numbers. You can reference the knowledge in KNOWLEDGE. Below are the numeric config options for your recommendations:
CONFIGS

A:   PHYSICAL_START (0x1000000)
      PHYSICAL_ALIGN (0x200000)

Figure 15: Instance of a `Value` type config option query

# B THEORETICAL PROOF

## B.1 PROOF OF PROPOSITION 1

**Proposition 1.** *Dual-layer knowledge graph resolves semantic issues by concept-instance mappings.*

**Proof:** Let $\mathcal{G} = (V, E)$ represent the dual-layer knowledge graph, where $V = \mathcal{E}_C \cup \mathcal{E}_I$ includes domain concepts $\mathcal{E}_C$ (high-level tuning objectives) and kernel instance entities $\mathcal{E}_I$ (config options). The edge set $E = \mathcal{R}_C \cup \mathcal{R}_I \cup \mathcal{L}$ captures semantic relationships $\mathcal{R}_C$, kernel config dependencies $\mathcal{R}_I$, and cross-layer links $\mathcal{L}$ between high-level concepts and config options.

We define the mapping function $\pi : \mathcal{E}_C \to \mathcal{E}_I$, where each concept $c_i \in \mathcal{E}_C$ is mapped to config options $o_j \in \mathcal{E}_I$ based on semantic strength:

$$\pi(c_i) = \{o_j \in \mathcal{E}_I : \sigma(c_i, o_j) \geq \delta\}$$

where $\sigma(c_i, o_j)$ measures semantic alignment, and $\delta$ is a threshold for selecting strong mappings.

We enforce valid mappings using probabilistic functions for dependency and constraint satisfaction:

$$P_\mathcal{D}(o_i, o_j) = \frac{\delta_{\mathcal{D}(o_i, o_j)}}{1 + \exp(-\alpha \cdot \mathcal{D}(o_i, o_j))}$$

$$P_\mathcal{C}(o_i, o_j) = \frac{\delta_{C(o_i, o_j)}}{1 + \exp(-\beta \cdot C(o_i, o_j))}$$

The overall validity of configuration $K = \{o_1, o_2, \ldots, o_n\}$ is given by:

$$P_{\text{valid}}(K) = \prod_{(o_i, o_j) \in K} (P_\mathcal{D}(o_i, o_j) \cdot P_\mathcal{C}(o_i, o_j))$$

This ensures the configuration satisfies both dependency and constraint relations.

The semantic strength between concept $c_i$ and config option $o_j$ is:

$$\sigma(c_i, o_j) = \frac{e^{-\|\vec{c_i} - \vec{o_j}\|^2}}{1 + e^{-\|\vec{c_i} - \vec{o_j}\|^2}}$$

where $\vec{c_i}$ and $\vec{o_j}$ are their respective vector embeddings, and $\|\vec{c_i} - \vec{o_j}\|^2$ is the squared Euclidean distance.

Finally, the valid mapping set is:

$$\mathcal{L} = \{(c_i, o_j) : \sigma(c_i, o_j) \geq \delta \text{ and } P_{\text{valid}}(K) > \tau\}$$

where $\tau$ is a threshold for the overall configuration validity.

In summary, the dual-layer knowledge graph bridges the semantic gap between high-level objectives and low-level config options, ensuring efficient and accurate kernel tuning via semantic alignment and probabilistic validation.

## B.2 PROOF OF PROPOSITION 2

**Proposition 2.** *Knowledge-driven reasoning over KG mitigates LLM hallucinations in kernel tuning.*

**Proof:** Let the high-level tuning objective $q$ be represented by a set $E_q = \{e_1, e_2, \ldots, e_n\}$, where each $e_i \in E_q$ corresponds to an abstract tuning goal. We map each entity $e_i$ to kernel config options using the mapping function $\phi : E_q \to E_C$, combining pattern matching ($\psi_{PM}$) and LLM-based semantic matching ($\psi_{LLM}$):

$$\phi(e_i) = \begin{cases} \psi_{PM}(e_i), & \text{if } \psi_{PM}(e_i) \neq \emptyset, \\ \psi_{LLM}(e_i), & \text{otherwise.} \end{cases}$$

Next, reasoning over the OD-KG explores paths $\pi(e_s)$ from a tuning concept $e_s \in E_C$ to config options $c_i \in E_I$, with each path's relevance computed as:

$$\rho(\pi(e_s)) = \prod_{i=1}^{n} \sigma(r_i) \cdot \omega(e_i),$$

where $\sigma(r_i)$ measures the semantic strength of relations and $\omega(e_i)$ captures the contextual importance of entities.

These functions are defined as:

$$\sigma(r_i) = \frac{1}{1 + e^{-\alpha \cdot d(r_i)}}, \quad \omega(e_i) = \frac{1}{1 + e^{-\beta \cdot h(e_i)}},$$

where $d(r_i)$ quantifies the semantic dissimilarity and $h(e_i)$ represents entity importance, with $\alpha$ and $\beta$ controlling sensitivity.

The relevance threshold $\tau$ filters out weakly related paths, defining the valid set of kernel configurations $K_q$ as:

$$K_q = \{e_i \in E_I \mid \rho(\pi(e_s)) \geq \tau\}.$$

This pruning minimizes hallucinations by excluding irrelevant configurations, ensuring that the reasoning process remains accurate and grounded in semantic consistency.

In summary, knowledge-driven reasoning over the KG enables precise kernel tuning by linking high-level tuning goals to concrete configurations, minimizing hallucinations and improving the robustness of LLM-based kernel tuning systems.

### B.3   PROOF OF PROPOSITION 3

**Proposition3.** Continuous knowledge maintenance ensures tuning accuracy and robustness.

**Proof:**   To prevent inaccuracy and invalid tuning due to kernel iteration, we employ continuous updates to the knowledge graph. Let $S(t) = (O(t), E(t), C(t))$ denote the kernel space at version $t$, and $S(t + 1) = (O(t + 1), E(t + 1), C(t + 1))$ represent the updated kernel space at version $t + 1$. The challenge lies in ensuring that the knowledge graph is incrementally updated to reflect changes in the kernel space, while retaining the semantic integrity of the prior version.

We define the config option delta $\Delta O_{add}$ as the set of newly added config options and $\Delta O_{del}$ as the set of deprecated options:

$$\Delta O_{add} = \{o \in O(t + 1) \mid o \notin O(t)\}, \quad \Delta O_{del} = \{o \in O(t) \mid o \notin O(t + 1)\}.$$

For options that persist across versions but exhibit changes in their domain definitions or dependency relations, we re-parse and update their corresponding entities and edges in the knowledge graph.

We update the knowledge graph by incorporating newly added options, removing deprecated options, and adjusting the mappings of existing options. The update rule for the kernel space is as follows:

$$G_I(t + 1) = G_I(t) \cup \Delta E_{add} \cup \Delta E_{mod}, \quad G_C(t + 1) = G_C(t).$$

This ensures that all new config options are considered and that deprecated or outdated information is removed, reducing the risk of hallucinations due to outdated kernel knowledge.

We further reduce hallucinations by ensuring that all config options are consistently grounded in the most up-to-date, relevant knowledge. The cross-layer mapping function is updated as:

$$L(t + 1) = \{(e_I, \text{related to}, e_C) \mid e_I \in \Delta E_{add}, e_C \in E_C(t + 1)\}.$$

By maintaining semantic consistency through structured knowledge interaction, we ensure that the reasoning remains robust and aligned with the latest kernel configurations, thereby mitigating hallucinations during the kernel tuning process. Additionally, this approach ensures that the generated kernel configurations remain contextually grounded and semantically relevant, thus overcoming the limitations of traditional LLM-based methods.

## C  KNOWOS ALGORITHM DETAILS

**Algorithm Overview.**  Algorithm 1 describes the core procedure for generating kernel configurations in a knowledge-driven manner using the KnowOS framework. It consists of two primary stages: heuristic value assignment and performance-aware refinement.

---

**Algorithm 1** Knowledge-driven Configuration Generation in KnowOS

---

1: **Input:** Candidate config options $K_q$, OD-KG $\mathcal{G}$, aligned concepts $\mathcal{E}_C^q$
2: **Output:** Valid kernel configuration $K$
3: **Step 1: Heuristic Inference for Option Value Assignment.**
4: Initialize $K \leftarrow \emptyset$
5: **repeat**
6:     Identify candidate configuration set $K_t$ from $K_q$
7:     $K_q \leftarrow K_q \setminus K_t$
8:     **for** each config option $o_t \in K_t$ **do**
9:         $x_t \leftarrow \texttt{LLM\_Infer}(o_t \mid \mathcal{E}_C^q, \mathcal{G}, K_t)$
10:         **if** $\texttt{Valid}(K_t \cup (o_t, x_t)) = \texttt{False}$ **then**
11:             Prune current assignment.
12:         **else**
13:             Add $(o_t, x_t)$ to $K_t$: $K_t = K_t \cup (o_t, x_t)$
14:         **end if**
15:     **end for**
16: **until** $K_q = \emptyset$
17: **Step 2: Performance-aware Final Configuration Generation.**
18: **for** each $(o_t, x_t) \in K$ **do**
19:     $x_t^* \leftarrow \arg \max\limits_{x \in \mathcal{D}_{o_t}} P(K \cup \{(o_t, x)\}, q)$
20:     **if** $\texttt{IsValid}(K \cup \{(o_t, x_t^*)\})$ **then**
21:         $K \leftarrow (K \setminus \{(o_t, x_t)\}) \cup \{(o_t, x_t^*)\}$
22:     **end if**
23: **end for**
24: **return** $K$

---

**Step 1: Heuristic Inference for Option Value Assignment.**  Given a set of candidate configuration options $K_q$, the algorithm iteratively selects subsets $K_t$ and attempts to infer suitable values for each option $o_t \in K_t$ using an LLM-based inference mechanism. The inference is conditioned on three inputs: the aligned tuning concepts $\mathcal{E}_C^q$, the OD-KG $\mathcal{G}$, and the current partial configuration $K_t$. After value inference, the resulting assignment $(o_t, x_t)$ is validated against kernel constraints and dependency rules. If valid, the assignment is retained; otherwise, the path is pruned. This phase continues until all candidate options have been processed.

**Step 2: Performance-aware Final Configuration Generation.**  Once a valid configuration $K$ is obtained, the algorithm optionally refines it by optimizing each option's value to maximize a performance objective $P(K, q)$. For each $(o_t, x_t) \in K$, the algorithm searches within the domain $\mathcal{D}_{o_t}$ for a value $x_t^*$ that yields the highest estimated performance, provided the updated configuration remains valid. The refined value is then used to update the configuration. This step ensures that the final configuration not only adheres to structural correctness but also maximizes utility under the given objective.

**Outcome.**  The algorithm returns a valid and performance-aligned configuration $K$ that maps high-level objectives to low-level kernel options through structured reasoning and LLM guidance.

## D  BENCHMARK DETAILS

In our experiments, we employed the following five distinct benchmarks to measure the performance differences among kernel configurations generated by various methods:

**UnixBench.** UnixBench Byte UnixBench Developers (1983) is an open-source benchmarking tool for Unix-like operating systems (such as Linux and BSD) that measures system performance across CPU, memory, and file I/O operations.

**LEBench.** LEBench Ren et al. (2019) is a microbenchmark suite that measures the performance of the 13 kernel operations that most significantly impact a variety of popular applications.

**RedisBench.** RedisBench Sanfilippo (2009) is a command-line utility included with Redis for measuring the performance of a Redis server by simulating multiple clients performing actions on the server.

**ApacheBench.** ApacheBench Apache Software Foundation (1997) is a command-line tool designed for benchmarking and load testing HTTP web servers.

**Sysbench.** Sysbench Akopytov (2004) is a popular, open-source, and modular benchmarking tool primarily used to test the performance of database servers and other system components like CPU, memory, and file I/O.

## E  EVALUATION DETAILS

### E.1  SETUP

Our experimental setups are shown in Table 4.

Table 4: The details of four representative Linux distributions. We used Ubuntu 22.04, Fedora 41, Debian 12 and openEuler 22.03 as the experiment environment for overall kernel performance test4.2.

| OS | Version | Kernel | Main Scenario |
|---|---|---|---|
| Ubuntu | 22.04 | Linux 6.2.16 | Desktop, Server, IoT |
| Fedora | 41 | Linux 6.2.16 | Development & Test |
| Debian | 12 | Linux 6.1.45 | Embedded System |
| openEuler | 22.03 | Linux 6.6.45 | Cloud Computing, AI |

### E.2  EMBEDDED BOARD EVALUATION

Additionally, we also conduct experiments on an embedded development board equipped with the SiFive Unmatched U740 system-on-chip, which features a multi-core, 64-bit dual-issue, superscalar RISC-V processor. We generated 8 configurations for Fedora using KnowOS and AutoOS separately, and compiled them into a kernel to run UnixBench. We selected the two best results from all the results of KnowOS and AutoOS as the final results, as shown in Table 5. As can be seen, KnowOS and AutoOS achieved total score improvements of 25.6% and 23.2%, respectively, with a small gap between the two in terms of total scores. This is due to the fact that we haven't added the RISC-V kernel knowledge to the knowledge base yet, so KnowOS lacks knowledge on how to improve kernel performance.

Table 5: Board test

| | Dhrystone | Whetstone | Execl Throughput | File Copy 1024 | File Copy 256 | File Copy 4096 | Pipe Throughput | Context Switching | Process Creation | Shell Scripts 1 | Shell Scripts 8 | System Call | Total Score |
|---|---|---|---|---|---|---|---|---|---|---|---|---|---|
| Default | **559** | 201 | 174 | 187 | 196 | 198 | 152 | 66 | 91 | 256 | 636 | 366 | 211 |
| AutoOS | 555 | 198 | **247** | 241 | **317** | **217** | **204** | 129 | 127 | 240 | 617 | **429** | 260 (+23.2%) |
| KnowOS | 552 | **202** | 247 | **242** | 307 | 211 | 201 | **141** | **130** | **259** | **675** | 429 | **265 (+25.6%)** |

### E.3 INFLUENCE OF DIFFERENT PROMPTS

We employed the following five distinct descriptive approaches to characterize our optimization objectives in order to validate the impact of different descriptive methods on optimization outcomes.

**P1.** I want to improve the performance of Redis.

**P2.** Fine-tune Redis for better performance.

**P3.** I would like to enhance the efficiency of Redis.

**P4.** Boost the performance of Redis.

**P5.** My goal is to increase Redis performance.

We ran ApacheBench on these five generated configs, and the results are shown in Table 6

Table 6: ApacheBench score of different prompts

| Score (ops/sec)\Prompt | P1 | P2 | P3 | P4 | P5 |
|---|---|---|---|---|---|
| **KnowOS** | 189377.98 | 189350.24 | 189370.56 | 189355.20 | 189382.10 |
| **w/o KG** | 155827.86 | 155801.54 | 155827.29 | 155815.60 | 155845.11 |

### E.4 TUNING COST OF KNOWOS

**Knowledge Graph Initialization.** The initial construction of the Knowledge Graph is a one-time cost. Initialization consumes approximately 1,100,000 tokens, requiring 12–18 minutes and costing about 5$.

**Tuning Cost.** A single optimization session consumes approximately 240,000 tokens, taking 10 to 20 minutes and costing about 1.2$.

**Knowledge Graph Maintenance Cost.** Updating the knowledge graph consumes approximately 80,000 tokens, requiring 5 to 12 minutes and costing about 0.4$.

## F LIMITATIONS AND FUTURE WORK

**Choice of OS.** We chose Linux for our experiments primarily due to its **open-source** nature and the rich configurations available via the **Kconfig** system. While other popular operating systems like Windows and macOS are closed-source, which prevent us from customization, and we are not yet aware of the existence of a structured form of configuration mechanism in these OSes similar to Linux. Additionally, we are unable to modify them because of copyright restrictions.

**Generalization to Non-Linux Kernels.** While our current work focuses on Linux, we are actively exploring the potential of extending KnowOS to other operating systems. We believe that the structured knowledge graph OD-KG can play a pivotal role in tackling similar challenges in these systems, provided the necessary configuration data becomes available. We are committed to exploring these avenues in future research.

## G THE USE OF LARGE LANGUAGE MODELS

We used LLMs to assist in refining the clarity and coherence of the writing in the paper. The LLMs were specifically employed to improve phrasing, ensure academic rigor, and enhance overall readability. Their contribution was strictly in the writing process, and all content was thoroughly reviewed and finalized by the authors.

