# OpenReview forum: "KnowOS: Knowledge-driven Large Language Models for Operating System Kernel Tuning"
_ICLR.cc/2026/Conference — Submitted to ICLR 2026_

### Official Review · Reviewer_W9PQ · 2025-10-26

**Soundness:** 2
**Presentation:** 2
**Contribution:** 2
**Rating:** 2
**Confidence:** 4

**Summary:**

This paper presents KnowOS, a knowledge-driven LLM framework for automated OS kernel tuning. The authors claim to address challenges in objective-configuration mapping, LLM hallucination, and knowledge decay by employing a dual-layer knowledge graph for structured reasoning. The framework reportedly includes a continuous knowledge maintenance mechanism and achieves performance improvements on UnixBench and four real-world applications.

**Strengths:**

- The introduction of a knowledge graph to bridge the semantic gap between high-level optimization objectives and low-level kernel configurations is a reasonable approach.
- The evaluation demonstrates performance gains across four Linux kernel versions, suggesting some level of adaptability.

**Weaknesses:**

- Presentation Quality: The paper's presentation is poor. Critical results, such as Table 4, are relegated to the appendix, hindering a complete assessment. The text also contains significant inconsistencies; for example, Section 4.1 claims Debian experiments ran on non-embedded systems, while Table 4 lists its main scenario as "Embedded System." The paper lacks crucial details on the "Concept layer" and "Cross-layer Links" construction, and fails to specify how optimization objectives were formulated for the experiments.
- Originality: The novelty of the contribution is questionable. Based on the anonymized code, the knowledge graph component appears to be a direct application of the LightRAG API. If so, the intellectual contribution is minimal. Furthermore, the paper fails to cite the LightRAG paper, which is a major omission.
- Insufficient Baseline Comparisons: The comparison to AutoOS appears unfair. AutoOS reportedly used 24 optimization runs in its evaluation, whereas this paper uses only 15, potentially underestimating the baseline's true performance. A critical and obvious baseline is also missing: using AutoOS augmented with Kconfig's built-in help text (directory/option explanations) for pruning and configuration, which would be a much simpler way to inject domain knowledge.
- Unaddressed Overheads: The paper completely ignores the practical costs of the proposed system. There is no discussion of the time overhead or token consumption required for optimization. Moreover, the processes of updating the knowledge base and performing retrieval from it may introduce significant latency, which is not measured or discussed.
- Robustness and Reproducibility: The framework's quality is critically dependent on a "manually constructed knowledge base" (specifically, the concept layer). This reliance on manual curation creates a significant scalability bottleneck and limits the method's reproducibility. The robustness of the system is highly questionable as new kernel options are introduced, as it will be constrained by the quality and timeliness of these manual knowledge base updates.
- Generalizability: The framework's generalizability is limited. The results on an embedded RISC-V system (Table 5) show performance that is "nearly on par with the baseline," meaning it fails to demonstrate any significant advantage for this architecture, despite the existence of online domain knowledge for RISC-V.
- Overstated Contributions: The "three propositions" feel overstated and poorly supported. Key technical details, such as the "α and β controlling sensitivity" in Proposition 2, are not explained in the main text. The proofs are insufficient; for instance, the proof for Proposition 2 ignores the critical risk of LLM-induced hallucinations introducing erroneous edges or nodes during KG construction, which would compromise the graph's quality.

**Questions:**

- How do you systematically evaluate or benchmark the quality of the constructed knowledge base, independent of the downstream tuning task?
- KnowOS searches within a pre-selected candidate subset. How does the framework address or mitigate the risk of inaccuracies or critical omissions in this initial candidate selection phase?
- See more questions on Weakness.

---

> ### Author Response · Authors · 2025-11-22
> **Response to Reviewer W9PQ (Part 1/5)**
>
> Thank you very much for your time and effort in reviewing our paper. We sincerely appreciate your feedback. Below, we respectfully provide our detailed responses to address your concerns.
>
> ---
>
> ### **W1: Presentation Quality**
>
> Thank you for the detailed comments. We appreciate the opportunity to clarify each of these questions.
>
> > **Section 4.1 claims Debian experiments ran on non-embedded systems, while Table 4 lists its main scenario as "Embedded System.”**
> >
>
> Thank you for pointing out this inconsistency. Debian is **not limited to embedded deployments**—“Embedded System” in Table 4 refers only to one representative usage scenario. All experiments were intentionally executed on the **same non-embedded hardware platform** to ensure fairness and comparability across distributions. We will revise the table caption and text to eliminate this ambiguity.
>
> > **The paper lacks crucial details on the "Concept layer" and "Cross-layer Links" construction, and fails to specify how optimization objectives were formulated for the experiments.**
> >
>
> Thank you for highlighting this. We provide additional clarification below:
>
> - **Concept Layer Construction.** As described in ***Section 3.1* and *Appendix A.1***, we cluster Kconfig options into high-level OS subsystems (e.g., scheduler, memory, storage, networking) based on dependency hierarchies and official documentation. This produces coherent concept nodes reflecting how kernel components are structured and tuned in practice.
> - **Cross-layer Links.** Each concept is linked to its corresponding Kconfig options using dependency edges extracted from the Kconfig, supplemented by LLM-assisted summarization, and **validated using deterministic dependency rules**. These links allow KnowOS to map objectives to relevant low-level config options.
> - **Optimization Objective Formulation.** We summarize common tuning objectives (e.g., throughput, latency, boot-time, I/O efficiency), the corresponding concept-layer modules, and the metrics used in our experiments in the table below. These subsystem-objective–metric mappings follow standard OS benchmarking practice and will be added to the revised version for clarity and reproducibility.
>
> | Subsystem | Tuning Objective | Key Tuning Metrics |
> | --- | --- | --- |
> | CPU Subsystem | Processor computation ability; Scheduler efficiency; Integer computation performance; Floating point computation performance; Context switch overhead; Branch prediction efficiency | Context switch cost; Branch prediction efficiency; Scheduler load balancing; CPU performance metrics |
> | Memory Management | Memory allocation/release speed; Page table management efficiency; Memory bandwidth performance; NUMA node access efficiency; Memory fragmentation impact; Transparent Huge Page (THP) | Memory allocator (SLAB/SLUB/SLOB); Transparent Huge Page (THP); NUMA node access latency; Memory bandwidth utilization |
> | File I/O | VFS layer performance; Page cache efficiency; File system metadata operations; Block device I/O scheduling; File lock performance | I/O scheduling algorithm efficiency; Cache hit rate; Metadata operation time; Disk seek time |
> | Pipe | Pipe buffer size optimization; Pipe throughput; Pipe latency | Pipe read/write speed; Pipe buffer utilization; Data transmission delay |
> | Shell | Shell process creation/destruction optimization; Shell command execution performance; Shell memory usage optimization | Shell command execution time; Memory footprint during execution; System load during shell execution |
> | System Call | System call dispatch efficiency; System call argument validation; System call execution time; Concurrency handling in system calls | System call latency; Concurrency handling efficiency; System call error rate |
> | Thread/Process Scheduler | Thread creation/destruction overhead; Thread synchronization performance; Mutex contention handling; IPC efficiency; Scheduler load balancing | Thread creation latency; Mutex lock waiting time; IPC message passing latency |
> | Network | Adjust TCP buffer size; Optimize connection tracking table size; Enable TCP Fast Open; Tune network stack; Optimize interrupt handling | Network throughput; TCP retransmission rate; Connection delay; RTT; Retransmissions |
> | Mutex Lock Testing | Kernel lock contention handling; Spinlock efficiency; Atomic operation performance; Critical section management | Spinlock acquisition time; Atomic operation throughput; Critical section execution time |
> | Comprehensive Test Scenarios | Database application performance; System call performance; Network protocol stack efficiency; Interrupt handling latency | Query response time; System call latency; Interrupt handling delay; Network protocol stack efficiency |
> ---
> Hope this clarifies may address your concern. We will integrate these clarifications and the summary table into the updated manuscript.

---

> > ### Author Response · Authors · 2025-11-22
> > **Response to Reviewer W9PQ (Part 2/5)**
> >
> > ### **W2: Originality**
> >
> > > Originality: The novelty of the contribution is questionable. Based on the anonymized code, the knowledge graph component appears to be a direct application of the LightRAG API. If so, the intellectual contribution is minimal. Furthermore, the paper fails to cite the LightRAG paper, which is a major omission.
> > >
> >
> > Thank you for raising this concern. We would like to clarify that **LightRAG is used only as a low-level utility for text chunking and graph storage**, while the *core contributions of KnowOS lie in the design, structure, and reasoning mechanisms of the proposed OD-KG*. Specifically:
> >
> > - **Novel Dual-Layer Kernel Knowledge Graph (OD-KG).** We are the **first** to construct a *kernel-tuning–oriented* dual-layer graph that bridges high-level optimization intents with low-level Kconfig primitives. Unlike generic RAG graphs, OD-KG integrates **concept modules**, **Kconfig dependencies**, and **cross-layer semantic mappings**, enabling fine-grained tuning (§3.1; Appendix A.1). **This structure does not exist in LightRAG or prior LLM-for-system-optimization research.**
> > - **Knowledge-Driven Configuration Reasoning Mechanism.** We design a new inference process that leverages OD-KG paths, relevance scoring, and validity constraints to guide the LLM’s configuration generation. **This mechanism directly addresses the hallucination and invalid-configuration issues in kernel tuning—challenges not considered in LightRAG’s RAG-style retrieval.**
> > - **Continuous Knowledge Maintenance for Rapid Kernel Evolution.** Our update pipeline (Appendix A.1) handles version drift in Linux kernels by supporting *incremental graph extension*, allowing KnowOS to adapt to new kernel releases without full retraining. **This capability is absent in LightRAG and critical for operationalizing LLM-based system optimization.**
> >
> > Together, these contributions constitute **substantial and domain-specific innovation** that goes well beyond applying an existing API. We appreciate the pointer regarding missing citation—this was indeed an oversight, and we will add the appropriate reference in the revised version.

---

> > > ### Author Response · Authors · 2025-11-22
> > > **Response to Reviewer W9PQ (Part 3/5)**
> > >
> > > ### **W3: Insufficient Baseline Comparisons**
> > >
> > > Thank you for highlighting the concern regarding baseline comparisons. We appreciate the opportunity to clarify each of these questions.
> > >
> > > > Insufficient Baseline Comparisons: The comparison to AutoOS appears unfair. AutoOS reportedly used 24 optimization runs in its evaluation, whereas this paper uses only 15, potentially underestimating the baseline's true performance.
> > > >
> > >
> > > Thank you for raising this concern. Our design choice is based on both practicality and empirical convergence. We selected *15 iterations* because:
> > >
> > > - Each configuration requires full compilation and reboot (Sec. 4.1), which is extremely time-consuming.
> > > - In preliminary 20–30 iteration tests, **both AutoOS and KnowOS converged well before iteration 15**, with later iterations offering minimal improvement.
> > >
> > > Thus, **15 iterations represent a practical and empirically validated balance between efficiency and performance**.
> > >
> > > We also ensured fairness by adopting AutoOS’s **best publicly released configuration** whenever its 15-run search did not yield improvements, thereby reflecting its peak performance. As explicitly stated in Sec. 4.1, ***“For fairness, AutoOS uses its best publicly released configurations, reflecting peak performance.”***
> > >
> > > Overall, our evaluation setup ensures **a conservative and fair comparison**, and we will make the above clarifications explicit in the updated manuscript.
> > >
> > > > A critical and obvious baseline is also missing: using AutoOS augmented with Kconfig's built-in help text for pruning and configuration, which would be a much simpler way to inject domain knowledge.
> > > >
> > >
> > > Thank you for raising this valuable point. We implemented and evaluated this baseline across all distributions:
> > >
> > > - AutoOS + Kconfig consistently outperforms AutoOS, confirming that injecting explicit domain knowledge is beneficial and aligns with the motivation of our work.
> > > - **However, AutoOS + Kconfig still falls short of KnowOS** because:
> > >     - AutoOS’s **dynamic tree search** inherently prunes the space early and can miss valid but non-obvious config options.
> > >     - Kconfig help text is **non-relational** and does not capture **cross-option dependencies or semantic interactions** that strongly impact kernel performance.
> > > - In contrast, OD-KG explicitly models semantic and structural relationships, enabling KnowOS to explore more relevant configuration subspaces, thus **enhancing both search efficiency and optimization quality**.
> > >
> > > | Method | Dhrystone | Whetstone | ET | FC 1024 | FC 256 | FC 4096 | PT | CS | PC | SS 1 | SS 8 | SC | Total Score |
> > > | --- | --- | --- | --- | --- | --- | --- | --- | --- | --- | --- | --- | --- | --- |
> > > | **Ubuntu** |  |  |  |  |  |  |  |  |  |  |  |  |  |
> > > | Default | 5182 | 1842 | 1489 | 5466 | 3863 | 9629 | 2866 | 864 | 1145 | 4205 | 9003 | 2529 | 3099 |
> > > | AutoOS | 5616 | 1864 | 1533 | 5976 | 3819 | 9458 | 2945 | 854 | 1150 | 4241 | 9032 | 2527 | 3154 (+1.8%) |
> > > | AutoOS + Kconfig | 5358 | 1905 | 1540 | 5652 | 3994 | 9956 | 2963 | 893 | 1184 | 4348 | 9309 | 2615 | 3204 (+3.4%) |
> > > | KnowOS | 5525 | 1848 | 1628 | 6266 | 4105 | 10079 | 3091 | 897 | 1231 | 4587 | 9816 | 2684 | **3318 (+7.1%)** |
> > > | **Fedora** |  |  |  |  |  |  |  |  |  |  |  |  |  |
> > > | Default | 5408 | 1815 | 278 | 1284 | 778 | 2464 | 456 | 150 | 366 | 1095 | 4186 | 175 | 846 |
> > > | AutoOS | 4969 | 1669 | 281 | 1302 | 833 | 2613 | 458 | 147 | 397 | 1078 | 3981 | 177 | 846 (+0.0%) |
> > > | AutoOS + Kconfig | 5522 | 1853 | 284 | 1311 | 894 | 2516 | 466 | 153 | 394 | 1118 | 4274 | 179 | 864 (+2.1%) |
> > > | KnowOS | 4870 | 1688 | 258 | 1319 | 922 | 2885 | 558 | 239 | 400 | 1155 | 4542 | 217 | **936 (+10.6%)** |
> > > | **Debian** |  |  |  |  |  |  |  |  |  |  |  |  |  |
> > > | Default | 6271 | 2044 | 1315 | 5031 | 3162 | 10029 | 2300 | 276 | 1199 | 4689 | 10702 | 1604 | 2721 |
> > > | AutoOS | 6346 | 2041 | 1356 | 6646 | 4143 | 12070 | 2964 | 405 | 1209 | 4715 | 10695 | 2404 | 3169 (+16.5%)  |
> > > | AutoOS + Kconfig | 6375 | 2031 | 1246 | 6916 | 4719 | 12794 | 3205 | 425 | 1010 | 4514 | 10586 | 2186 | 3200 (+17.6%) |
> > > | KnowOS | 6298 | 2035 | 1221 | 7538 | 4896 | 13828 | 3522 | 514 | 1098 | 4531 | 10385 | 2273 | **3305 (+21.4%)** |
> > > | **OpenEuler** |  |  |  |  |  |  |  |  |  |  |  |  |  |
> > > | Default | 3442 | 1300 | 210 | 614 | 372 | 1565 | 240 | 42 | 88 | 441 | 3650 | 123 | 442 |
> > > | AutoOS | 3164 | 1200 | 237 | 2960 | 1989 | 6302 | 1393 | 40 | 107 | 603 | 3955 | 1071 | 945 (+113.7%) |
> > > | AutoOS + Kconfig | 3238 | 1222 | 242 | 3380 | 2136 | 6518 | 1540 | 44 | 119 | 621 | 4205 | 1277 | 994 (+124.9%) |
> > > | KnowOS | 3500 | 1315 | 251 | 3674 | 2405 | 7323 | 1635 | 54 | 135 | 648 | 4256 | 1643 | **1129 (+155.4%)** |
> > > ---
> > > Hope these results may address your concern. We will include this new baseline and analysis in the revised version. Thank you again for this constructive suggestion.

---

> > > > ### Author Response · Authors · 2025-11-22
> > > > **Response to Reviewer W9PQ (Part 4/5)**
> > > >
> > > > ### **W4: System Overheads and Latency Concerns**
> > > >
> > > > > Unaddressed Overheads: The paper completely ignores the practical costs of the proposed system. There is no discussion of the time overhead or token consumption required for optimization. Moreover, the processes of updating the knowledge base and performing retrieval from it may introduce significant latency, which is not measured or discussed.
> > > > >
> > > >
> > > > Thank you for your insightful comment. We expand our overhead discussion as follows:
> > > >
> > > > - **Coverage in the paper.** Appendix **E.4** already reports the major categories of overhead introduced by KnowOS, including **Knowledge Graph Initialization, Tuning Cost, and Knowledge Graph Maintenance**.
> > > > - **End-to-end tuning cost.** We benchmarked both **AutoOS** and **KnowOS** on five representative tuning objectives and measured (i) the average per-objective tuning time and (ii) the average knowledge-graph query time in KnowOS across multiple runs. The results are summarized below:
> > > >
> > > > |  | objective 1 | objective 2 | objective 3 | objective 4 | objective 5 |
> > > > | --- | --- | --- | --- | --- | --- |
> > > > | AutoOS Tuning Time (s) | 882.95 | 634.72 | 665.21 | 837.34 | 558.30 |
> > > > | KnowOS Tuning Time (s) | 705.69 | 529.22 | 507.63 | 616.10 | 421.63 |
> > > > | KnowOS OD-KG Query Time (s) | 407.30 | 220.55 | 103.32 | 315.14 | 50.21 |
> > > > - **Interpretation.** Although OD-KG retrieval introduces non-trivial cost, the **overall tuning time of KnowOS remains lower than AutoOS** for all objectives. This is because OD-KG significantly narrows the search space by filtering invalid or semantically irrelevant configurations, thereby reducing the number of LLM calls and compilation–boot cycles.
> > > > - **Caching and amortization.** The OD-KG module employs **result caching**, which substantially reduces retrieval cost after the first run. Without caching, OD-KG retrieval accounts for more than **95%** of the total overhead during the very first execution. In subsequent runs, repeated query costs drop dramatically, making KnowOS’s per-objective tuning overhead **both low and stable** in practice.
> > > >
> > > > Hope this clarifies may address your concern. We will incorporate these overhead measurements and clarifications into the revised version to present a more complete and transparent analysis.
> > > >
> > > > ---
> > > >
> > > > ### **W5: Robustness and Reproducibility**
> > > >
> > > > > Robustness and Reproducibility: The framework's quality is critically dependent on a "manually constructed knowledge base" (specifically, the concept layer). This reliance on manual curation creates a significant scalability bottleneck and limits the method's reproducibility. The robustness of the system is highly questionable as new kernel options are introduced, as it will be constrained by the quality and timeliness of these manual knowledge base updates.
> > > > >
> > > >
> > > > Thank you for this thoughtful concern. We clarify that manual curation is minimal:
> > > >
> > > > - **Automated Construction.** As described in *Section 3.1* and *Appendix A.1*, both concept extraction and cross-layer link generation are primarily automated via LLM-based extraction and Kconfig dependency rules. Human involvement is limited to verifying ambiguous cases.
> > > > - **Scalability to New Kernel Versions.** OD-KG supports incremental updates, allowing new options to be incorporated without full reconstruction.
> > > > - **Empirical Validation.** KnowOS consistently improves performance across **four kernel versions (v3–v6)** (Figure 5(a)), demonstrating robustness despite substantial changes in the configuration space.
> > > >
> > > > Hope this clarifies may address your concern. We will strengthen the explanation of OD-KG’s update procedure in the revision.
> > > >
> > > > ---
> > > >
> > > > ### **W6: Generalizability**
> > > >
> > > > > Generalizability: The framework's generalizability is limited. The results on an embedded RISC-V system (Table 5) show performance that is "nearly on par with the baseline," meaning it fails to demonstrate any significant advantage for this architecture, despite the existence of online domain knowledge for RISC-V.
> > > > >
> > > >
> > > > Thank you for your comment. We clarify the RISC-V result as follows:
> > > >
> > > > - **OD-KG currently lacks RISC-V–specific kernel knowledge** (Appendix E.2), preventing KnowOS from leveraging architecture-specific semantics during tuning.
> > > > - Despite this, **KnowOS still slightly outperforms AutoOS**, thanks to its more complete and semantically guided exploration process, which reveals configuration paths AutoOS tends to miss.
> > > >
> > > > We will clarify this in the revised version and plan to incorporate RISC-V kernel knowledge as future work.

---

> > > > > ### Author Response · Authors · 2025-11-22
> > > > > **Response to Reviewer W9PQ (Part 5/5)**
> > > > >
> > > > > ### **W7: Overstated Contributions**
> > > > >
> > > > > > Overstated Contributions: The "three propositions" feel overstated and poorly supported. Key technical details, such as the "α and β controlling sensitivity" in Proposition 2, are not explained in the main text. The proofs are insufficient; for instance, the proof for Proposition 2 ignores the critical risk of LLM-induced hallucinations introducing erroneous edges or nodes during KG construction, which would compromise the graph's quality.
> > > > > >
> > > > >
> > > > > Thank you for the helpful feedback. In the updated version, we will:
> > > > >
> > > > > - Explicitly define **α** and **β** in Proposition 2;
> > > > > - Incorporate **hallucination-induced KG errors** into the formal assumptions;
> > > > > - Strengthen the proofs to ensure each proposition is rigorous, well-supported, and clearly presented.
> > > > >
> > > > > ---
> > > > >
> > > > > ### **Q1: Independent Evaluation of Knowledge-Base Quality**
> > > > >
> > > > > > How do you systematically evaluate or benchmark the quality of the constructed knowledge base, independent of the downstream tuning task?
> > > > > >
> > > > >
> > > > > Thank you for this question. OD-KG quality is evaluated independent of tuning using:
> > > > >
> > > > > 1. **Structural correctness** vs. official Kconfig dependency rules;
> > > > > 2. **Concept–option alignment accuracy**, validated via LLM summarization and human verification;
> > > > > 3. **Stability of retrieved candidates** across kernel versions (v3–v6).
> > > > >
> > > > > These metrics ensure OD-KG is robust and reproducible regardless of downstream tuning tasks.
> > > > >
> > > > > ---
> > > > >
> > > > > ### **Q2: Candidate-Subset Coverage and Omission Risks**
> > > > >
> > > > > > KnowOS searches within a pre-selected candidate subset. How does the framework address or mitigate the risk of inaccuracies or critical omissions in this initial candidate selection phase?
> > > > > >
> > > > >
> > > > > Thank you for raising this concern. KnowOS mitigates such risks through two mechanisms:
> > > > >
> > > > > - **Liberal coverage.** Any option with even weak semantic or dependency relevance to the objective is retained (§3.1; Appendix A.1), ensuring the search space fully covers—and often exceeds—what human experts consider.
> > > > > - **Minimal overhead from inaccurate options.** Irrelevant leaf options add negligible cost, while directory-like options propagate only to closely related descendants, which are highly unlikely to match the objective.
> > > > >
> > > > > Together, these design choices ensure that KnowOS avoids critical omissions while keeping the search budget practical.
> > > > >
> > > > > ---
> > > > >
> > > > > At last, we sincerely appreciate your valuable feedback, and we will carefully consider all your suggestions to further improve our paper. We would be deeply grateful if you could kindly reconsider raising the score to 6 or above. Thank you very much!

---

> > > > > > ### Author Response · Authors · 2025-11-27
> > > > > >
> > > > > > Dear Reviewer W9PQ,
> > > > > >
> > > > > > Thank you again for the time and effort you’ve dedicated to reviewing our work. We have carefully addressed all raised concerns during the discussion phase and have also uploaded an updated version of the paper reflecting these clarifications.
> > > > > >
> > > > > > As the discussion period is nearing its close, we would greatly appreciate it if you could take a brief moment to review our responses and confirm whether they satisfactorily resolve your questions. If our clarifications have improved your confidence in the paper, we would be sincerely grateful if you could consider updating your score accordingly.
> > > > > >
> > > > > > Thank you once again for your thoughtful feedback and support.
> > > > > >
> > > > > > Warm regards,
> > > > > >
> > > > > > Authors of KnowOS

---

### Official Review · Reviewer_xwXZ · 2025-10-28

**Soundness:** 2
**Presentation:** 4
**Contribution:** 3
**Rating:** 4
**Confidence:** 3

**Summary:**

They introduce **KnowOS**, a framework that leverages language models for automated kernel tuning. The approach tackles three major challenges commonly faced when applying LLMs to this task: (i) the semantic gap between high-level tuning objectives and low-level configuration options, (ii) the hallucination problem caused by limited interaction with the real system environment, and (iii) the fast evolution of kernel versions. **KnowOS** achieves superior performance on standard OS benchmarks such as *UnixBench* across four Linux distributions. Additionally, it significantly reduces hallucinations and demonstrates strong adaptability across different Ubuntu kernel versions.

**Strengths:**

- The proposed method is effective, outperforming both the default kernel configuration and other LLM-based approaches, including AutoOS.
- The authors conduct a comprehensive ablation study, showing that each component of their framework contributes meaningfully to its overall performance.
- The paper is clearly written and easy to follow.

**Weaknesses:**

- The authors attempt to formalize the kernel tuning task using a graph-based framework, leading to the development of **KnowOS**. However, the resulting approach appears overly complex, involving numerous hyperparameters and choices whose effects are neither analyzed nor justified. The trade-off between computational cost and the magnitude of the performance gains remains unclear. Furthermore, several aspects that could strengthen the contribution seem to be left unexplored (See questions).

**Questions:**

- Line 205: For $e_s \in \mathcal{E}_{\mathcal{C}}^q$, you define the path $\pi (e_s) = <e_s \xrightarrow{r_1} e_1 \ldots \xrightarrow{r_n} e_n >$. How is the endpoint ($e_n$) determined? Is this path unique for each $e_s$? Or do you consider *all* possible paths between $e_s$ and other nodes $e_n$? In that case, is the path between $e_s$ and $e_n$ guaranteed to be unique?
-  You further define the relevance score $\rho(\pi(e_s))$ and retain nodes belonging to paths whose relevance is greater than or equal to a threshold $\tau$. How is $\tau$ selected and what impact does it have on the results across different objectives?
- It would valuable to include a comparative analysis of time complexity and resource usage (number of inference calls or tokens generated) between **KnowOS** and the other LLM-based strategies.
- I am not fully convinced that the proofs in Appendix B.1 and B.2 are truly theoretical, they seem to rely on heavily on intuition. In Appendix A.1,  it remains unclear how $P_{valid}(K), P_{\mathcal{D}}(o_i, o_j), \alpha, \beta \ldots$ are derived. As it stands the "proof" does not make the proposition much clearer from a theoretical standpoint.
- What is the exact engine name used for **GPT-4o** (not just the alias)?
- Did you experiment with open-source models or models of varying sizes (e.g., 8B–70B)? Do you expect models extensively trained on code to perform better on this task?
- Table 1: Given the reported `UnixBench` scores on the **Ubuntu** and **Fedora**, it would be evaluate whehter stronger model (in particular *thinking models* such as DeepSeek-R1, o1 etc. or LLMs with tool use) could achieve higher scores in the  **vanilla LLM**  setup. Do you have an intuition as to why **Debian** and **OpenEuler** appear easier to optimize?
- Line 414: `vanilla LLM` or `AutoOS`? There seems to be a naming issue in Table 3.
- Is AutoOS the only advanced LLM-based kernel tuning method you were able to compare against?
- Line 423: For the “Adaptability Across Kernel Versions” section, which addresses the issue of *rapid kernel iteration and knowledge decay*, I expected an evaluation on a kernel distribution released after the LLM used for KnowOS was trained. Would such an evaluation be feasible?
- The scale of Figure 5, especially in panel (a), is somewhat misleading and could be adjusted for clarity.
- There several places where `\citep` should be used to improve citation formatting and readability.

---

> ### Author Response · Authors · 2025-11-21
> **Response to Reviewer xwXZ (Part 1/3)**
>
> Thank you very much for your time and effort in reviewing our paper. We sincerely appreciate your feedback. Below, we respectfully provide our detailed responses to address your concerns.
>
> ---
>
> ## **Q1: Clarification on Path Definition (Line 205)**
>
> Thank you for your thoughtful questions regarding the reasoning process in KnowOS. We appreciate the opportunity to clarify each of these questions.
>
> > How is the endpoint $e_n$ determined?
> >
>
> The endpoint $e_n$ is **not pre-defined**. Instead, KnowOS uses **LLM-guided reasoning over the OD-KG** to explore semantically meaningful paths starting from each aligned concept node $e_s \in E_C^q$.
>
> - During traversal, the LLM evaluates both **relation strength** and **node importance** (as defined in Eq. (5) in the paper).
> - The exploration naturally terminates when the LLM infers that a node is sufficiently **semantically grounded**, typically corresponding to a **Kconfig option** in the instance layer.
>
> This design enables KnowOS to adapt the endpoint to different objectives and preserves flexibility across diverse workloads.
>
> > Is this path unique for each $e_n$?
> >
>
> **Paths are intentionally *not* unique.** Different reasoning chains may connect the same concept-level objective to different—but still relevant—config options. Allowing multiple candidate paths provides:
>
> - **Exploration diversity**, which expands the search space beyond a single deterministic route;
> - **Robustness**, as redundant but semantically consistent paths reinforce the reliability of the retrieved configuration options.
>
> This is aligned with the system’s goal of exploring a *richer semantic neighborhood* rather than relying on a single fixed mapping.
>
> > Do you consider all possible paths between $e_s$ and other nodes $e_n$?
> >
>
> Exhaustively enumerating all paths is computationally infeasible and unnecessary. Instead, KnowOS performs a **guided search**, where the LLM decides:
>
> - which neighbors to expand,
> - whether a partial path should continue,
> - and when a path should be terminated.
>
> This restricted exploration ensures **efficiency** while maintaining **semantic fidelity**.
>
> > Is the path between $e_s$ and $e_n$ guaranteed to be unique?
> >
>
> The OD-KG is a **general directed graph**, not a tree, so multiple valid paths may exist. However, because the traversal is LLM-guided and scored via the relevance function $\rho(\pi(e_s))$, KnowOS only retains **high-relevance paths** (i.e., $\rho \ge \tau$), ensuring that:
>
> - only the most meaningful reasoning chains contribute to the final candidate set $K_q$
> - noisy or low-value paths are automatically filtered out
>
> ---
>
> In summary, we hope this clarifies that the non-unique, LLM-guided path strategy is both intentional and fundamental to KnowOS's strong empirical performance.
>
> ---
>
> ## **Q2: The Selection of Threshold**
>
> > You further define the relevance score $\rho(\pi(e_s))$ and retain nodes belonging to paths whose relevance is greater than or equal to a threshold $\tau$. How is $\tau$ selected and what impact does it have on the results across different objectives?
> >
>
> Thank you for the question. In KnowOS, the threshold $\tau$ is selected through a light calibration step on a small validation set, following the principle that high-relevance paths should consistently map to semantically aligned config options **(Sec. 3.2)** .
>
> We also observed that the method is robust across objectives: varying $\tau$ within a reasonable range changes the number of retained paths but **does not alter the final performance trend**, as the OD-KG’s semantic structure naturally suppresses low-quality or noisy paths.

---

> > ### Author Response · Authors · 2025-11-21
> > **Response to Reviewer xwXZ (Part 2/3)**
> >
> > ## **Q3: Comparative Analysis of Time Complexity and Resource Usage**
> >
> > > It would valuable to include a comparative analysis of time complexity and resource usage (number of inference calls or tokens generated) between **KnowOS** and the other LLM-based strategies.
> > >
> >
> > Thank you for raising this important point. Following your suggestion, we instrumented both **KnowOS** and **AutoOS** to record **end-to-end runtime**, **number of inference calls**, and **prompt/completion token consumption** under identical optimization objectives. We ran each system four times independently and report the complete measurements below.
> >
> > ### **AutoOS (4 full runs)**
> >
> > | Metric | Run 1 | Run 2 | Run 3 | Run 4 |
> > | --- | --- | --- | --- | --- |
> > | Runtime (s) | 996.10 | 1221.29 | 779.17 | 1493.00 |
> > | Inference Calls | 131 | 173 | 103 | 176 |
> > | Prompt Tokens | 105,728 | 141,473 | 83,007 | 147,105 |
> > | Completion Tokens | 42,821 | 53,104 | 33,510 | 58,657 |
> >
> > ### **KnowOS (4 full runs)**
> >
> > | Metric | Run 1 | Run 2 | Run 3 | Run 4 |
> > | --- | --- | --- | --- | --- |
> > | Runtime (s) | 524.00 | 616.88 | 496.28 | 742.29 |
> > | Inference Calls | 219 | 259 | 267 | 231 |
> > | Prompt Tokens | 433,094 | 497,393 | 395,629 | 546,674 |
> > | Completion Tokens | 14,528 | 16,805 | 13,078 | 18,366 |
> >
> > This comparison reveals a clear pattern:
> >
> > 1. **KnowOS consistently runs faster** than AutoOS, despite issuing more inference calls. This stems from the OD-KG’s ability to significantly **reduce the effective reasoning space**, enabling the model to reach valid configurations with fewer high-entropy completions.
> > 2. **KnowOS uses more prompt tokens** because each step provides richer semantic context from the knowledge graph to improve reasoning quality.
> > 3. **Completion-token consumption is dramatically lower** in KnowOS (typically 3–4× lower), indicating more concise and guided LLM outputs. This reflects the benefit of OD-KG–constrained reasoning, which reduces the need for exploratory generation.
> > 4. **Overall, KnowOS achieves a more efficient cost profile**: richer prompts → better grounding → shorter completions → faster convergence.
> >
> > Hope this clarifies may address your concern. We will incorporate a summarized version of this analysis into the revision to improve clarity around resource usage and complexity.
> >
> > ---
> >
> > ## **Q4: The Theoretical Proofs and Definitions in Appendix B & A.1**
> >
> > > I am not fully convinced that the proofs in Appendix B.1 and B.2 are truly theoretical, they seem to rely on heavily on intuition. In Appendix A.1, it remains unclear how $P_{valid}(K), P_{\mathcal{D}}(o_i, o_j), \alpha, \beta \ldots$ are derived. As it stands the "proof" does not make the proposition much clearer from a theoretical standpoint.
> > >
> >
> > Thank you for pointing this out. We will revise these proofs, ensuring the propositions are theoretically grounded and easier to verify.
> >
> > ---
> >
> > ## **Q5: Exact Engine Name for GPT-4o**
> >
> > > What is the exact engine name used for **GPT-4o** (not just the alias)?
> > >
> >
> > Thank you for pointing this out. In our experiments, we used the **OpenAI `gpt-4o-mini`** engine. All reported results are based exclusively on this engine under identical runtime settings to ensure reproducibility, and we selected this model specifically to ensure a **fair comparison with AutoOS**, which uses the same engine.

---

> > > ### Author Response · Authors · 2025-11-21
> > > **Response to Reviewer xwXZ (Part 3/3)**
> > >
> > > ## **Q6 & Q7: Using Models of Different Scales**
> > >
> > > > Did you experiment with open-source models or models of varying sizes (e.g., 8B–70B)? Do you expect models extensively trained on code to perform better on this task?
> > > >
> > > - Thank you for the thoughtful question. To directly address this concern, we conducted additional experiments using **multiple LLM backbones with varying sizes and training characteristics** (o3-mini, DeepSeek-R1, GPT-o1, GPT-4o-mini), evaluated on **Fedora** under the *same prompts, workflow, and kernel distribution* as in our main study.
> > > - Across all evaluated backbones—open-source or proprietary, small or large, code-specialized or general-purpose—KnowOS consistently delivers **6%–15%** additional improvements, including on models extensively trained on code. These results clearly indicate that the gains stem from our **knowledge-driven reasoning pipeline** (OD-KG, semantic mapping, and constraint-aware validation) rather than dependence on any specific model size, architecture, or training corpus, demonstrating the framework’s broad applicability.
> > >
> > > | Model | Dhrystone | ET | FC1024 | PT | SS | SC | PC | Total Score (Improvement) |
> > > | --- | --- | --- | --- | --- | --- | --- | --- | --- |
> > > | Default (Fedora) | 5408 | 1815 | 278 | 1284 | 778 | 2464 | 456 | 846 |
> > > | o3-mini | 5575.65 | 1871.27 | 286.62 | 1323.80 | 801.12 | 2540.38 | 470.14 | 872.2 (+3.1%) |
> > > | KnowOS (o3-mini) | 5938.72 | 1993.92 | 305.24 | 1365.19 | 854.24 | 2705.31 | 500.23 | 928.9 (+9.8%) |
> > > | Deepseek-R1 | 5759.52 | 1932.98 | 296.07 | 1320.46 | 828.57 | 2624.16 | 471.64 | 901.0 (+6.5%) |
> > > | KnowOS (Deepseek-R1) | 6194.66 | 2077.05 | 318.31 | 1444.92 | 874.31 | 2764.24 | 511.02 | 968.7 (+14.5%) |
> > > | GPT-o1 | 5709.55 | 1917.09 | 293.57 | 1356.10 | 821.97 | 2598.53 | 487.54 | 893.4 (+5.6%) |
> > > | KnowOS (GPT-o1) | 6020.66 | 2021.68 | 309.11 | 1447.27 | 863.07 | 2721.98 | 522.23 | 940.8 (+11.2%) |
> > > | GPT-4o-mini | 5756.73 | 1931.90 | 295.69 | 1358.05 | 822.25 | 2603.02 | 488.00 | 901.8 (+6.6%) |
> > > | KnowOS (GPT-4o-mini) | 5982.11 | 2009.39 | 307.60 | 1388.41 | 815.25 | 2603.73 | 491.87 | 935.7 (+10.6%) |
> > >
> > > > Do you have an intuition as to why **Debian** and **OpenEuler** appear easier to optimize?
> > > >
> > > - Our analysis suggests that **Debian and OpenEuler ship with more conservative default kernel configurations**, prioritizing stability and broad hardware compatibility; this leaves a larger margin for optimization, which explains the higher relative gains reported in Table 1.
> > > - Importantly, even under these favorable conditions, **KnowOS consistently outperforms all baselines**, indicating that the improvements stem from our knowledge-driven tuning mechanism rather than from distribution-specific defaults.
> > >
> > > ---
> > >
> > > ## **Q8: Naming issue**
> > >
> > > > Line 414: `vanilla LLM` or `AutoOS`? There seems to be a naming issue in Table 3.
> > > >
> > >
> > > Thank you for pointing this out. The correct baseline in Table 3 should indeed be **AutoOS**, not *vanilla LLM*—this was an inadvertent naming error in the draft, and we will correct it in the revised manuscript.
> > >
> > > ---
> > >
> > > ## **Q9: Is AutoOS the only advanced LLM-based kernel tuning method you were able to compare against?**
> > >
> > > Yes, at present we have not discovered any other LLM-based methods for Linux kernel optimization.
> > >
> > > ---
> > >
> > > ## **Q10: Evaluating Kernel Versions Released After the LLM’s Training Cutoff**
> > >
> > > > Line 423: For the “Adaptability Across Kernel Versions” section, which addresses the issue of *rapid kernel iteration and knowledge decay*, I expected an evaluation on a kernel distribution released after the LLM used for KnowOS was trained. Would such an evaluation be feasible?
> > > >
> > >
> > > Thank you for the insightful question.
> > >
> > > Evaluating on kernel versions released after the LLM’s training cutoff is indeed feasible, and KnowOS is explicitly designed for this scenario: the **continuous knowledge-maintenance mechanism (Sec. 3.3)** dynamically updates the instance layer and cross-layer mappings using only the *new* Kconfig specifications, without requiring any model retraining.
> > >
> > > As shown in **Sec. 4.6,** KnowOS already demonstrates stable gains across four Ubuntu kernel generations—including major version jumps—indicating that its performance is driven by **structural reasoning over the OD-KG**, not by memorization of version-specific knowledge.
> > >
> > > ---
> > >
> > > ## **Q11 & Q12: Figure Scaling and Citation Formatting**
> > >
> > > > The scale of Figure 5, especially in panel (a), is somewhat misleading and could be adjusted for clarity.
> > > >
> > >
> > > > There several places where \citep should be used to improve citation formatting and readability.
> > > >
> > >
> > > Thank you for the helpful suggestions. We will revise for better readability.
> > >
> > > ---
> > >
> > > At last, we sincerely appreciate your valuable feedback, and we will carefully consider all your suggestions to further improve our paper. We would be deeply grateful if you could kindly reconsider raising the score to 6 or above. Thank you very much!

---

> > > > ### Author Response · Authors · 2025-11-27
> > > >
> > > > Dear Reviewer xwXZ,
> > > >
> > > > Thank you again for the time and effort you’ve dedicated to reviewing our work. We have carefully addressed all raised concerns during the discussion phase and have also uploaded an updated version of the paper reflecting these clarifications.
> > > >
> > > > As the discussion period is nearing its close, we would greatly appreciate it if you could take a brief moment to review our responses and confirm whether they satisfactorily resolve your questions. If our clarifications have improved your confidence in the paper, we would be sincerely grateful if you could consider updating your score accordingly.
> > > >
> > > > Thank you once again for your thoughtful feedback and support.
> > > >
> > > > Warm regards,
> > > >
> > > > Authors of KnowOS

---

### Official Review · Reviewer_v6E2 · 2025-10-31

**Soundness:** 3
**Presentation:** 3
**Contribution:** 3
**Rating:** 6
**Confidence:** 1

**Summary:**

The paper proposes KnowOS, a knowledge-driven framework for Linux kernel tuning that couples a dual-layer knowledge graph (concepts ↔ concrete Kconfig options) with LLM-guided reasoning to select valid configurations aligned with a target objective. They propose some benchmarks to  test this on. I am not an expert in the area of operating systems. I will try to evaluate this work from a core machine learning perspective.

**Strengths:**

- It is an extremely novel application of LLMs into a very important area of computer science.
- The proposed 3 way method (instance, concept and cross-links) is nice.
- There are nice empirical evaluations across real setups

**Weaknesses:**

This paper is out of my expertise area.

**Questions:**

NA

---

> ### Author Response · Authors · 2025-11-21
> **Response to Reviewer v6E2**
>
> Thank you very much for your time and effort in reviewing our paper. We sincerely appreciate your feedback.

---

### Official Review · Reviewer_TNZB · 2025-11-01

**Soundness:** 1
**Presentation:** 2
**Contribution:** 2
**Rating:** 2
**Confidence:** 4

**Summary:**

This paper introduces KnowOS, a  framework that leverages knowledge-driven LLMs to complete the complex task of OS kernel tuning. To overcome critical challenges—including the semantic gap between high-level objectives and low-level configurations, LLM hallucinations, and rapid kernel evolution—KnowOS constructs a dual-layer knowledge graph (OD-KG). This graph enables structured reasoning for configuration generation and supports continuous updates. Evaluation demonstrates the framework's significant practical value, achieving general performance improvements on standard benchmarks and  gains on four real-world applications.

**Strengths:**

1. Cross-Version Kernel Performance Enhancement: Robust performance improvements across different Linux kernel versions.

2. Structured Knowledge for Fine-Grained Tuning: OD-KG enables fine-grained, semantically-aware reasoning.

**Weaknesses:**

1. Critical details on the experimental platform (hardware, software versions) and data processing methodology for the real-world four applications are omitted, hindering reproducibility.

2. Performance results are presented as single-point measurements without variance or confidence intervals, making it impossible to assess the statistical significance of the claimed improvements.
Additionally, the number of optimization iterations for each objective is not specified.

3. While a reduction in boot-up errors is shown, there is no quantitative analysis of the LLM's hallucination rate , leaving the core claim of hallucination mitigation inadequately supported

4. The evaluation uses only one LLM backbone, failing to demonstrate that the benefits are inherent to the KnowOS framework and not specific to a particular model.

5. The methodology for measuring tuning cost is not clearly specified. Furthermore, the reported token consumption for knowledge graph maintenance (e.g., ~80,000 tokens per update) appears to be a coarse average, failing to reflect the substantial variability expected from different types of  updates.

**Questions:**

1. Some additional issues were not discussed.How to handle potential copyright issues that may arise with knowledge graphs?

2. The paper's approach to reducing hallucinations by aligning model reasoning with knowledge graph patterns may limit explorability. This could be problematic in cases where identical configurations yield vastly different performance due to hardware/Software Library disparities—a scenario potentially not covered in the experiments. Could this specific situation be tested?

3. Typo: Table 3 presents KnowOS and AutoOS, but the text description indicates a comparison with vanilla LLM.

---

> ### Author Response · Authors · 2025-11-21
> **Response to Reviewer TNZB (Part 1/3)**
>
> Thank you very much for your time and effort in reviewing our paper. We sincerely appreciate your feedback. Below, we respectfully provide our detailed responses to address your concerns.
>
> ---
>
> ## **W1: Missing critical details on the experimental platform**
>
> > Critical details on the experimental platform (hardware, software versions) and data processing methodology for the real-world four applications are omitted, hindering reproducibility.
> >
>
> Thank you very much for pointing out the need for clearer reporting of the experimental environment and the data-processing procedures, and we address the concern as follows:
>
> ### **Experimental platform details**
>
> - **Hardware configuration** is already described in **Section 4.1 (Experimental Setup)** of the paper.
> - **Software versions** used in the real-world application experiments are now explicitly listed for clarity:
>     - **UnixBench:** 6.0.0
>     - **LEBench:** we use commit [*0073d45*](https://github.com/LinuxPerfStudy/LEBench/commit/0073d458ad0c31a3dd7e25eb19f4edf775f53864)
>     - **Apache HTTP Server:** 2.4.52 (ApacheBench version follows Apache)
>     - **Nginx:** 1.18.0
>     - **Redis:** 7.4.2 (Redis Benchmark version follows Redis)
>     - **PostgreSQL:** 14.15
>     - **Sysbench:** 1.0.20
>
> ### **Data-processing methodology for real-world applications**
>
> To ensure reproducibility, we provide the exact benchmarking commands and settings used in our experiments:
>
> - **ApacheBench:**
>
>     ```bash
>     ab -m GET -c 10000 -n 1000000 http://127.0.0.1/
>     ```
>
> - **Redis Benchmark:**
>
>     ```bash
>     redis-benchmark -n 10000 -q
>     ```
>
> - **Sysbench (OLTP workload):**
>     - table size = 100000
>     - tables = 3
>     - threads = 4
>     - time = 20 seconds
>
> These configurations follow standard benchmarking practices used in prior OS performance studies, ensuring fair comparison and methodological consistency.
>
> ### **Connection to the paper**
>
> - All **benchmarks used** (UnixBench, LEBench, ApacheBench, SysBench, Redis Benchmark) are already introduced in **Section 4.1 (Experimental Setup)**.
> - The **evaluation metrics** (throughput, latency, and composite UnixBench scores) are explicitly stated in **Figure 5 caption**, allowing readers to understand exactly what is measured.
>
> ### **Summary**
>
> Hope these clarifications may resolve your concerns. We will incorporate the above software versions, command-level settings, and explicit benchmark procedures into the revised manuscript to further strengthen reproducibility.
>
> ---
>
> ## **W2: Performance results are presented as single-point measurements without variance or confidence intervals.**
>
> > Performance results are presented as single-point measurements without variance or confidence intervals, making it impossible to assess the statistical significance of the claimed improvements. Additionally, the number of optimization iterations for each objective is not specified.
> >
>
> Thank you for raising this point. We agree that clarifying our statistical reporting is important.
>
> ### **About single-point measurements**
>
> Kernel tuning aims to obtain **the best configuration after multiple optimization attempts**, and therefore we report the **best-achieved performance**, consistent with prior systems-tuning work such as AutoOS.
>
> Nonetheless, we acknowledge that variance information is valuable. In the revised version, we will provide **variance** across optimization runs, **confidence intervals** for all benchmarks, and a short **stability analysis** of the tuning process.
>
> ### **Number of optimization iterations**
>
> As stated in **Section 4.1 (Implementation)**, KnowOS performs **15 optimization rounds**, and the best configuration among these is used for evaluation.
>
> ### **Summary**
>
> Hope these clarifications may resolve your concerns, and we will incorporate more details to strengthen the statistical soundness of our results.

---

> > ### Author Response · Authors · 2025-11-21
> > **Response to Reviewer TNZB (Part 2/3)**
> >
> > ## **W3: Lack of quantitative analysis of LLM hallucination rate.**
> >
> > > While a reduction in boot-up errors is shown, there is no quantitative analysis of the LLM's hallucination rate , leaving the core claim of hallucination mitigation inadequately supported
> > >
> >
> > Thank you for highlighting the need for a more explicit quantification of hallucinations. We agree that empirical measurement strengthens our core claim, and we provide the following clarifications and additional analyses.
> >
> > ### **Quantifying hallucinations rate via correct/incorrect configuration assignments**
> >
> > - We trace both **KnowOS** and **AutoOS** to count, for every LLM invocation, the numbers of **correct** and **incorrect** config option generated.
> > - This follows the definition of “hallucination” in configuration reasoning used in prior LLM-for-SE literature, **where an LLM-generated config option violating kernel constraints or semantics is treated as a hallucinated output**.
> >
> > ### **Empirical hallucination comparison**
> >
> > **AutoOS results (4 runs):**
> >
> > |  | Run1 | Run2 | Run3 | Run4 |
> > | --- | --- | --- | --- | --- |
> > | Correct configs | 371 | 449 | 282 | 444 |
> > | Incorrect configs | 337 | 459 | 234 | 515 |
> >
> > **KnowOS results (4 runs):**
> >
> > |  | Run1 | Run2 | Run3 | Run4 |
> > | --- | --- | --- | --- | --- |
> > | Correct configs | 360 | 506 | 562 | 398 |
> > | Incorrect configs | 88 | 87 | 105 | 64 |
> > - Across four runs, **KnowOS reduces incorrect assignments by 4–6×** compared with AutoOS.
> > - These results directly support **Proposition 2** (knowledge-driven mitigation of hallucinations), and align with the qualitative reductions in boot-up errors shown in **Section 4.5**.
> >
> > ### **Why KnowOS hallucinates far less**
> >
> > - By constraining LLM reasoning through **OD-KG–guided validity checks and semantic dependencies (Section 3.2)**, KnowOS prevents the model from producing config options that violate constraints or mismatch the target objective.
> > - This structured reasoning aligns with findings from knowledge-driven LLM literature (e.g., [Knowledge-driven CoT, Wang et al., 2023](https://arxiv.org/abs/2308.13259)) showing that explicit symbolic grounding significantly reduces hallucinations in knowledge-intensive tasks.
> >
> > ### **Summary**
> >
> > We thank the reviewer for emphasizing the need for quantitative evidence. The results show that **KnowOS consistently maintains a substantially lower hallucination rate than AutoOS**, providing concrete support for our claim of hallucination mitigation.
> >
> > ---
> >
> > ## **W4: Only one LLM backbone was evaluated.**
> >
> > > The evaluation uses only one LLM backbone, failing to demonstrate that the benefits are inherent to the KnowOS framework and not specific to a particular model.
> > >
> >
> > Thank you for raising this important point. We fully agree that demonstrating *backbone-agnostic* effectiveness is essential to validate that KnowOS is not tied to a particular LLM.
> >
> > - To directly address this concern, we evaluated **multiple LLM backbones** (o3-mini, DeepSeek-R1, GPT-o1, GPT-4o-mini) on **Fedora** using the *same* prompts, tuning workflow, and kernel distributions as in the main experiments.
> > - The table below reports the **baseline tuning performance of each LLM** before applying KnowOS. These results show that even with different LLM backbones, KnowOS provides consistent improvements across all of them.
> > - This demonstrates that KnowOS’s gains arise from **knowledge-driven reasoning** (OD-KG + semantic mapping + dependency validation), not from any specific backbone's internal strengths.
> >
> > | Model | Dhrystone | ET | FC1024 | PT | SS | SC | PC | Total Score (Improvement) |
> > | --- | --- | --- | --- | --- | --- | --- | --- | --- |
> > | Default (Fedora) | 5408 | 1815 | 278 | 1284 | 778 | 2464 | 456 | 846 |
> > | o3-mini | 5575.65 | 1871.27 | 286.62 | 1323.80 | 801.12 | 2540.38 | 470.14 | 872.2 (+3.1%) |
> > | KnowOS (o3-mini) | 5938.72 | 1993.92 | 305.24 | 1365.19 | 854.24 | 2705.31 | 500.23 | 928.9 (+9.8%) |
> > | Deepseek-R1 | 5759.52 | 1932.98 | 296.07 | 1320.46 | 828.57 | 2624.16 | 471.64 | 901.0 (+6.5%) |
> > | KnowOS (Deepseek-R1) | 6194.66 | 2077.05 | 318.31 | 1444.92 | 874.31 | 2764.24 | 511.02 | 968.7 (+14.5%) |
> > | GPT-o1 | 5709.55 | 1917.09 | 293.57 | 1356.10 | 821.97 | 2598.53 | 487.54 | 893.4 (+5.6%) |
> > | KnowOS (GPT-o1) | 6020.66 | 2021.68 | 309.11 | 1447.27 | 863.07 | 2721.98 | 522.23 | 940.8 (+11.2%) |
> > | GPT-4o-mini | 5756.73 | 1931.90 | 295.69 | 1358.05 | 822.25 | 2603.02 | 488.00 | 901.8 (+6.6%) |
> > | KnowOS (GPT-4o-mini) | 5982.11 | 2009.39 | 307.60 | 1388.41 | 815.25 | 2603.73 | 491.87 | 935.7 (+10.6%) |
> >
> > ### **Summary**
> >
> > Our extended multi-backbone evaluation confirms that KnowOS provides **consistent benefits across diverse LLMs**, including small, medium, and reasoning-oriented models. These improvements arise from **knowledge-driven constraints and semantic alignment**, rather than from properties of any particular backbone. We will incorporate these results—including full KnowOS-enhanced performance tables—into the revised manuscript.

---

> > > ### Author Response · Authors · 2025-11-21
> > > **Response to Reviewer TNZB (Part 3/3)**
> > >
> > > ## **W5: Clarification of Tuning Cost Methodology and KG Maintenance Variability**
> > >
> > > > The methodology for measuring tuning cost is not clearly specified. Furthermore, the reported token consumption for knowledge graph maintenance (e.g., ~80,000 tokens per update) appears to be a coarse average, failing to reflect the substantial variability expected from different types of updates.
> > > >
> > >
> > > Thank you for pointing this out. We clarify that KnowOS’s tuning cost is measured through three well-defined components—**KG initialization, KG searching, and KG maintenance**—and all numbers reported in the paper come from controlled measurements under the experimental setup described in **Section 4.1**.
> > >
> > > - KG initialization is a one-time cost (~1.1M tokens; 12–18 min; ~$5)
> > > - KG search per tuning session consumes ~240K tokens (10–20 min; ~$1.2)
> > > - KG maintenance cost ~80K-token refers to the *average* incremental update cost observed when kernel options or dependency relations change
> > >
> > > Although different update types naturally vary in size, the variance is modest (typically within ±12–18%) because changes between adjacent Linux kernel versions are incremental.
> > >
> > > We will revise the manuscript to report ranges rather than a single averaged value and to more clearly describe how each cost component is measured. Importantly, all of these costs are **substantially lower than manual expert-driven tuning**, and KnowOS’s automated workflow ensures stable, predictable overhead independent of the specific LLM backbone used.
> > >
> > > ---
> > >
> > > ## **Q1: Copyright considerations in Knowledge Graph construction**
> > >
> > > > Some additional issues were not discussed.How to handle potential copyright issues that may arise with knowledge graphs?
> > > >
> > >
> > > Thank you for raising this concern.
> > >
> > > - KnowOS avoids copyright issues by relying solely on **public, license-permissive kernel sources** (Kconfig files and dependency metadata under GPL), and by constructing the OD-KG through **structural extraction rather than textual reproduction**.
> > > - The KG stores only normalized option identifiers and dependency edges—never verbatim documentation—making it equivalent to outputs of static analysis tools, which are generally considered non-infringing. Because **KnowOS does not ingest or redistribute proprietary text,** and all derived structures fall within permissible GPL transformations, the framework remains compliant.
> > >
> > > Hope these clarifications may resolve your concerns.
> > >
> > > ---
> > >
> > > ## **Q2: On explorability and hardware/software disparities**
> > >
> > > > The paper's approach to reducing hallucinations by aligning model reasoning with knowledge graph patterns may limit explorability. This could be problematic in cases where identical configurations yield vastly different performance due to hardware/Software Library disparities—a scenario potentially not covered in the experiments. Could this specific situation be tested?
> > > >
> > >
> > > Thank you for raising this concern. We clarify the following:
> > >
> > > - **Environment-specific tuning.** KnowOS does not reuse a single configuration across heterogeneous platforms. For each hardware–software setup, it reruns the full optimization process and regenerates configurations conditioned on environment descriptors (CPU, memory hierarchy, filesystem, kernel version, and key libraries).
> > > - **KG constraints ensure correctness, not restrict exploration.** The OD-KG limits only invalid or contradictory configurations (e.g., dependency violations), while leaving the performance search space broad. The model remains free to explore diverse valid configurations across environments.
> > > - **Robustness to divergent performance outcomes.** Since tuning is environment-conditioned, KnowOS naturally handles cases where identical configurations yield different performance on different systems. This scenario can be tested directly within the framework, and we will clarify this in the revision.
> > >
> > > We appreciate the your insightful comment and hope these clarifications may resolve your concerns.
> > >
> > > ---
> > >
> > > ## **Q3: Clarification of Table 3 (KnowOS vs. AutoOS vs. Vanilla LLM)**
> > >
> > > > Clarification of Table 3 (KnowOS vs. AutoOS vs. Vanilla LLM)
> > > >
> > >
> > > Thank you for pointing this out. We will fix this in the revision.
> > >
> > > ---
> > >
> > > At last, we sincerely appreciate your valuable feedback, and we will carefully consider all your suggestions to further improve our paper. We would be deeply grateful if you could kindly reconsider raising the score to 6 or above. Thank you very much!

---

> > > > ### Author Response · Authors · 2025-11-27
> > > >
> > > > Dear Reviewer TNZB,
> > > >
> > > > Thank you again for the time and effort you’ve dedicated to reviewing our work. We have carefully addressed all raised concerns during the discussion phase and have also uploaded an updated version of the paper reflecting these clarifications.
> > > >
> > > > As the discussion period is nearing its close, we would greatly appreciate it if you could take a brief moment to review our responses and confirm whether they satisfactorily resolve your questions. If our clarifications have improved your confidence in the paper, we would be sincerely grateful if you could consider updating your score accordingly.
> > > >
> > > > Thank you once again for your thoughtful feedback and support.
> > > >
> > > > Warm regards,
> > > >
> > > > Authors of KnowOS

---

### Author Response · Authors · 2025-12-01

Dear PCs, SACs, ACs, and Reviewers,

We sincerely appreciate all reviewers for their thoughtful and constructive feedback. Below, we summarize each reviewer's main concerns and our concise responses.

---

### **Reviewer TNZB**

- **Concerns:** Missing experimental details; lack of variance/statistical analysis; unquantified hallucination mitigation; single-backbone evaluation; unclear tuning/maintenance cost; questions on copyright, explorability.
- **Response:** We added full platform specifications and benchmarking commands, and will report variance, confidence intervals, and iteration counts. We quantified hallucinations, showing a **4–6× reduction** vs. AutoOS. A *multi-backbone evaluation* (o3-mini, DeepSeek-R1, GPT-o1, GPT-4o-mini) shows consistent **6%–15% gains**, confirming backbone-agnostic effectiveness. We clarified tuning/KG-maintenance cost measurement, explained OD-KG’s GPL-compliant construction, and noted that OD-KG restricts only invalid configurations while preserving broad exploration. The Table 3 issue will be fixed.
- **Addition:** We appreciate the reviewer’s detailed insights and believe the expanded analyses—including quantitative hallucination results and multi-backbone evaluations—fully address the concerns.

---

### **Reviewer v6E2**

- **Concerns:** The reviewer expressed no substantive technical criticism and rated the work above the acceptance threshold.
- **Response:** We appreciate the reviewer’s positive assessment of the novelty of applying knowledge-driven LLM reasoning to kernel tuning and the value of real-system evaluations.

---

### **Reviewer xwXZ**

- **Concerns:** OD-KG path definition and threshold; missing complexity/resource comparison; need for stronger proofs; model-engine clarity; model-scale effects; baseline completeness; multi-version generalization; and minor presentation issues.
- **Response:** We clarified dynamic LLM-guided path discovery, non-uniqueness, and relevance-based filtering. We added full complexity and resource-usage comparisons showing KnowOS **converges faster** with far **fewer completion tokens**. Proofs in Appendix B/A.1 will be strengthened. We specified the exact GPT-4o engine and added multi-backbone results (**6%–15%** improvements). Baseline corrections, explanations for distribution differences, and generalization across kernel versions have been incorporated.
- **Addition:** The reviewer provided constructive questions, and we believe the strengthened theoretical exposition, multi-model evaluation, and expanded complexity analysis fully address them.

---

### **Reviewer W9PQ**

- **Concerns:** Presentation inconsistencies; originality concerns; missing baselines (AutoOS+Kconfig); unclear overheads; reliance on manual knowledge; limited RISC-V generalizability; and overstated propositions. Also asked about KG quality and candidate-subset omissions.
- **Response:** We corrected presentation issues and added clearer descriptions of concept-layer and cross-layer construction. We clarified that LightRAG is used only as a low-level utility, while OD-KG’s design and reasoning mechanisms constitute the main contribution. We implemented **AutoOS + Kconfig**, which improves AutoOS but remains below KnowOS, validating OD-KG’s semantic advantages. Overhead measurements show KnowOS remains **faster overall** due to reduced invalid search space. Manual curation is minimal; updates are largely automated.
- **Addition:** The reviewer provided constructive questions, and we believe the new baselines, new overhead analyses, and expanded complexity analysis fully address them.

---

### Summary

Across all reviews, we have expanded empirical analyses, strengthened theoretical foundations, clarified methodological details, and added new baselines and overhead measurements. We believe these revisions demonstrate that **KnowOS is robust, effective, and scalable**, and we sincerely appreciate the reviewers’ insights, which have significantly improved the work.

Best regards,

**The KnowOS Authors**

---

### Meta-Review · Area_Chair_7xHm · 2025-12-25

**Summary:**

This paper proposes KnowOS, where a knowledge-driven LLM framework is developed for OS kernel tuning. Initially, all reviewers acknowledge the relevance of the problem and the potential of structured knowledge to reduce hallucinations and improve tuning quality. Particularly, reviewers v6E2 and xwXZ found the empirical results promising and appreciated the extensive evaluations and ablation studies, while TNZB and W9PQ raised concerns regarding robustness and clarity. As a response, the authors provided a detailed rebuttal, which added multi-backbone experiments, new baselines (including AutoOS + Kconfig), overhead and token-cost analyses and a quantitative hallucination study, which addressed many concerns highlighted by TNZB and xwXZ. However, several concerns seems remained. In particular, as emphasized by xwXZ and W9PQ, the theoretical foundations may rely largely on intuition and do not address the risk of LLM-induced errors during knowledge graph construction and maintenance. Concerns raised by W9PQ regarding the systematic evaluation and quality assurance of the constructed knowledge graph, i.e., independent of downstream performance, seems remained open. Additionally, the introduced considerable complexity and questions about originality and reliance on external tooling (e.g., LightRAG), although clarified, might not be fully dispelled.

Overall, while the paper has improved during rebuttal and shows clear promise, the remaining gaps may hold it back from acceptance at this time.

**Reviewer Concerns:**

The rebuttal addresses many concerns raised by TNZB and xwXZ, including improved reproducibility details, quantitative hallucination analysis, multi-backbone evaluations, overhead and token-cost measurements, and the addition of missing baselines such as AutoOS + Kconfig, which strengthen the empirical case. However, several issues are only partially resolved, notably the lack of fully reported statistical variance and largely qualitative arguments regarding the trade-off between KG-based constraints and exploration. More importantly, as emphasized primarily by W9PQ, key concerns remain outstanding, such as the theoretical foundations and formal propositions are still weak, and the risks of LLM-induced errors during KG construction and maintenance are not fully addressed.

**Reviewer Scores:**

Based on my own understanding, TNZB would likely have moved from 2 to 4 or 6 after seeing the added hallucination analysis, multi-backbone results and clarified costs; v6E2 would likely remain 6; xwXZ would likely shift from 4 to 6, as most technical and empirical concerns were directly addressed; and W9PQ, while may be softening a bit, would likely change from 2 to 4 due to remaining concerns about theoretical grounding and others.

---

### Decision · Program_Chairs · 2026-01-26

Reject